# Central amygdala CRF+ neurons promote heightened threat reactivity following early life adversity in mice

Camila Demaestri [1], Margaux Pisciotta[2], Naira Altunkeser [3], Georgia Berry [4], Hannah Hyland [4], Jocelyn Breton[4,5], Anna Darling [3], Brenna Williams [6] & Kevin G. Bath[4,5] ✉

Failure to appropriately predict and titrate reactivity to threat is a core feature of fear and anxiety-related disorders and is common following early life adversity (ELA). A population of neurons in the lateral central amygdala (CeAL) expressing corticotropin releasing factor (CRF) have been proposed to be key in processing threat of different intensities to mediate active fear expression. Here, we use in vivo fiber photometry to show that ELA results in sex-specific changes in the activity of CeAL CRF+ neurons, yielding divergent mechanisms underlying the augmented startle in ELA mice, a translationally relevant behavior indicative of heightened threat reactivity and hypervigilance. Further, chemogenic inhibition of CeAL CRF+ neurons selectively diminishes startle and produces a long-lasting suppression of threat reactivity. These findings identify a mechanism for sex-differences in susceptibility for anxiety following ELA and have broad implications for understanding the neural circuitry that encodes and gates the behavioral expression of fear.

A debilitating feature of fear and anxiety-related disorders, such as generalized anxiety disorder (GAD) and post-traumatic stress disorder (PTSD), is excessive anticipation and response to threats. While the ability to predict and react to imminent danger is critical for survival, disproportionate anticipation of threat can manifest as pathology. A major contributor to anxiety disorders is early life adversity (ELA), increasing lifetime risk by 30%, with higher rates in women compared to men[1–4]. Clinical evidence has identified a critical influence of heightened amygdala activity towards fearful stimuli contributing to disrupted emotional regulation in children, adolescents, and adults following ELA[5–10]. Heightened behavioral and neural responding to fearful stimuli may represent a failure to appropriately regulate and titrate reactivity to real or perceived threat[7,11–13]. However, the mechanisms that gate excessive reactivity to threat and that increase susceptibility in ELA-exposed individuals remain poorly understood.

A broad array of research has described the involvement of the lateral central amygdala (CeAL) in fear and stress-induced anxiety by mediating a host of processes, including associative learning, valence processing, attention allocation, and orchestrating the somatic response to threat[14–18]. Recent interest has explored how the CeAL encodes and possesses the flexibility necessary for processing stimuli across a spectrum of threat-related emotional states[19,20]. Support for regulating behavioral responses to threats of varying degrees has been demonstrated in a subtype of CeAL neurons that express the modulatory neuropeptide corticotropin-releasing factor (CRF +). CeAL CRF + neurons have been shown to enhance associative and non-associative startle[21–25], to facilitate learning about weak threats[26], to regulate stress-induced anxiety-like behaviors[27–29], to process salience[30,31], and to regulate aspects of contextual and cued fear extinction[32–34]. The broad role of CeAL CRF + in stress and threat-

[1]Doctoral Program in Neurobiology and Behavior, Columbia University, New York, USA. [2]Department of Neuroscience and Behavior, Barnard College of Columbia University, New York, NY, USA. [3]Department of Neuroscience, Columbia University, New York, NY, USA. [4]Division of Developmental Neuroscience, Research Foundation for Mental Hygiene, New York State Psychiatric Institute, New York, NY, USA. [5]Department of Psychiatry, Columbia University Irving Medical Center, New York, NY, USA. [6]Doctoral Program in Cellular and Molecular Physiology & Biophysics, Columbia University, New York, NY, USA. ✉e-mail: Kevin.Bath@nyspi.columbia.edu

related contexts suggests that their function may act in an anticipatory capacity that depends on the perceived proximity or intensity of the threat and prior associative or non-associative experiences.

Pre-clinical and clinical studies have reported persistent changes in the function of CRF + caused by ELA that could significantly impact how individuals process and respond to threat[35–37], including driving an augmented startle reflex[11–13,38–40]. Enhanced startle has been used as a diagnostic symptom for GAD, PTSD, and panic disorder and has been associated with elevated levels of CRF[41–46]. Further, the startle response has gained validity as a behavioral indicator of stress-induced anxiety and states elicited by imminent threat because it reflects the body's automatic and involuntary response to a threat. Heightened startle response in humans and rodents indicates increased arousal, hyper-vigilance, and emotional reactivity associated with perceived danger[39,43,47]. Despite extensive work characterizing the startle pathway and implicating CRF + neurons and signaling in the CeAL in startle, there remains a significant gap in understanding the role of CRF + neurons in generating startle in the context of differing proximity to threat and whether ELA maybe disrupting CRF + signaling to induce greater threat reactivity.

Here, we tested the effects of the limited bedding and nesting (LBN) model of ELA in mice on startle response in adulthood. To assess threat reactivity during different threat proximities, we measured startle response to a white noise (WN) that either co-terminated with a fear-conditioned tone (CS +) or occurred in the absence of the CS + (noise alone; NA), thus assessing startle when the proximity of threat is high (CS + trials) and ambiguous (NA trials). We then tested whether CeAL CRF + neurons were differentially activated as a function of ELA and sex using c-Fos and in vivo fiber photometry. Lastly, we tested the necessity of CeAL CRF + neuron activity for gating startle and the impact of CRF + neuron inhibition on long-term threat and fear expression. This line of research offers a distinct perspective on understanding the heterogeneity of threat responding and holds promise for understanding the neurobiological basis of sex differences in risk and for guiding personalized treatment strategies that address the underlying causes of fear and anxiety-related disorders to manage specific symptoms.

## Results

### ELA enhanced startle in the context of both high and ambiguous proximity of threat

Adult male and female WT mice reared in control (Ctrl), or ELA conditions underwent cued-fear conditioning to associate a shock (0.5 mA, 0.5 sec) with a tone (12 kH, 70 dB, 30 sec). Roughly twenty-four hours later, the startle reflex to a white noise stimulus (WN) at pseudorandom intensities 95 dB, 100 dB, and 105 dB (50 msec) was measured to determine the impact of ELA on threat reactivity in males and females (Fig. 1a). The startle paradigm was subdivided into two Blocks. In Block 1, startle was measured to a series of 9 WN stimuli in the absence of the CS + (noise alone; NA), providing an assessment of threat generalized to the context in which the timing of impending threat is ambiguous. In Block 2, startle was measured to a pseudo-random series of 18 WN stimuli: 9 co-terminated with the CS +, and 9 NA trials. Startle during the CS + trials assesses threat reactivity during high proximity of threat. Startle during the NA trials in Block 2 provides an assessment of threat reactivity during ambiguous threats, given that the proximity (timing) of the WN was unpredictable (pseudorandom and long intertrial intervals of 60–120 sec). Further, levels of cue-potentiated startle were measured by dividing the startle to the CS+ by the startle to the NA in Block 1 (CS + / NA), which assesses the startle potentiated by the CS + .

ELA rearing significantly augmented startle during NA trials in Block 1, and in Block 2 both during CS + and NA trials, indicating a broad effect of ELA on threat reactivity (Fig. 1b, c). Notably, in a separate group of WT mice, the enhanced startle phenotype in ELA

mice was not evident prior to any experimental manipulation (Supplementary Fig. 1a). These findings suggest that ELA may differentially impact startle response during basal conditions compared to startle elicited by threat stimuli[12,13]. ELA-enhancement in startle was also not due to a stronger acquisition of the CS +, as the rate and magnitude of freezing during CS+ conditioning did not significantly differ by rearing condition (Supplementary Fig. 1B, C). Further, ELA rearing did not alter cue-potentiated startle (Fig. 1d). Together, these findings indicate that ELA induced generalized hypersensitivity to the startling WN exhibited as exaggerated startle in the context of both high and ambiguous threat (NA and CS + trials) without altering CS + acquisition or startle potentiated solely by the fear cue.

### ELA rearing enhanced startle and resulted in the preferential recruitment of CeAL CRF + neurons in female ELA mice

To investigate the role of CeAL CRF + neurons in gating startle and whether the activity was altered as a function of ELA, we collected timed-perfused brains from adult CRF-ires-Cre::Ai14tdTomato mice bred from CRF-ires-Cre mice crossed with Ai14tdTomato mice (CRF x Ai14tdT) one-hour following startle testing. We first used in situ hybridization to confirm the co-expression of endogenous *Crf* transcript with tdTomato (tdT) transcript in Cre + mice and that *Crf* density levels in Cre+ mice (relative to DAPI) were comparable to those observed in Cre- mice (Supplementary Fig. 2A–D). Thus, the presence of Cre under the *Crf* promoter did not impact *Crf* cell density. Furthermore, we counted the density of CRF + neurons using the tdT reporter in control and ELA mice (Supplementary Fig. 2E). We did not find effects of ELA on CRF + neuron density, suggesting that the presence of the reporter did not contribute to changes in tdT + cells following ELA. Lastly, we confirmed that the ELA-induced startle phenotype was present in the CRF x Ai14[tdT] mouse line (Supplementary Fig. 2H). These data together validate the use of this transgenic line for investigating ELA and sex effects on the density of CRF-expressing neurons in the CeA.

Next, we quantified the density of c-Fos + neurons relative to DAPI. ELA reared mice showed increased c-Fos density in the CeAL, suggesting greater overall recruitment of the CeAL following ELA (Fig. 1e, f). To determine whether the CRF + neuron population was differentially recruited as a consequence of ELA, the percent of CRF + neurons that were c-Fos + and the percent of c-Fos + neurons that were CRF + was quantified. We found that the percent of CRF + neurons that were c-Fos + was greater in female ELA mice compared to female Ctrls, an effect not observed in males, indicating a sex-dependent impact of ELA on the proportion of CRF + neurons that were recruited during startle (Fig. 1g). Importantly, the density of CeAL CRF + neurons did not differ by sex or rearing, indicating a functional rather than structural change (Supplementary Fig. 2E). The percent of c-Fos + cells that co-expressed CRF was numerically higher in ELA-reared females. However, the sex by rearing interaction did not reach significance (Fig. 1h). These results are suggestive of possible biased recruitment of CRF + neurons in ELA females compared to other recruited cell types. Interestingly, this measure was positively correlated with startle (Fig. 1l). Thus, biased recruitment of CeAL CRF + neurons may predict heightened startle reactivity.

In a separate cohort of CRF x Ai14[tdT] mice, we collected timed-perfused brains following fear conditioning to test whether female ELA mice similarly recruited a greater proportion of CRF + neurons during cued-fear acquisition. We did not observe a bias towards CRF + cell recruitment in ELA female mice following fear acquisition, suggesting that increased activity of CRF + neurons following ELA in females derives from their engagement during threat recall and not during cued-fear acquisition (Supplementary Fig. 2G, H). Interestingly, female mice exhibited increased c-Fos + expression and a greater percentage of c-Fos + cells that were CRF + (Supplementary Fig. 2G, H). We speculate that the sex-specific bias to drive greater

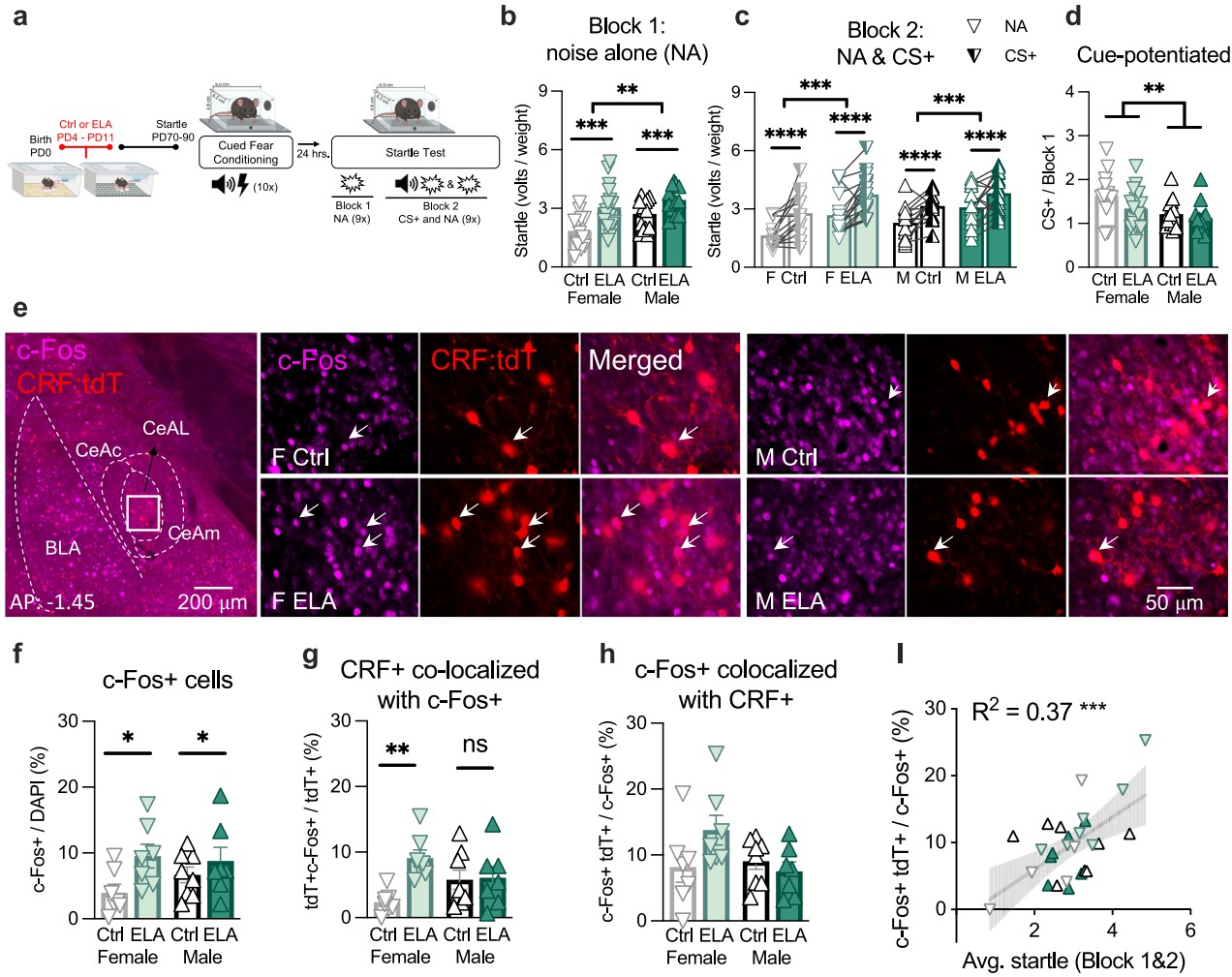

**Fig. 1 | ELA rearing enhanced startle and resulted in the preferential recruitment of CeAL CRF + neurons in female ELA mice. a** Experimental design. Female (F), male (M), control (Ctrl), and ELA (ELA) reared mice underwent the startle task in adulthood. **b** Startle in Block 1 was enhanced by ELA and sex (2-way-ANOVA $_{rearing}$ $F_{(1, 53)} = 16.34$, $p = 0.0002$, $\eta^2 = 0.55$; 2-way-ANOVA $_{sex}$ $F_{(1, 53)} = 6.92$, $p = 0.01$, $\eta^2 = 0.36$) and not an interaction (2-way-ANOVA $_{sex \times rearing}$ $F_{(1,53)} = 0.98$, $p = 0.32$, $\eta^2 = 0.13$). $n = 13$ (F Ctrl), 16 (F ELA), 15 (M Ctrl), 16 (M ELA). These data have been replicated in three independent experiments. **c** Startle was greater in CS + trials and was enhanced by ELA (3-way-RM-ANOVA $_{cue}$ $F_{(1, 53)} = 50.67$, $p = 2.9e{-}9$, $\eta^2 = 0.97$; 3-way-RM-ANOVA $_{rearing}$ $F_{(1, 53)} = 15.86$, $p = 0.0002$, $\eta^2 = 0.87$). Startle did not differ by sex or an interaction (3-way-ANOVA $_{sex}$ $F_{(1,53)} = 2.78$, $p = 0.10$, $\eta^2 = 0.36$; 3-way-RM-ANOVA $_{sex \times rearing}$ $F_{(1,53)} = 0.51$, $p = 0.47$, $\eta^2 = 0.14$). $n = 13$(F Ctrl), 16 (F ELA), 15 (M Ctrl), 16 (M ELA). These data have been replicated in three independent experiments. **d** Cue-potentiated startle was heightened in females (2-way-ANOVA $_{sex}$ $F_{(1, 53)} = 8.25$, $p = 0.005$, $\eta^2 = 0.39$). $n = 13$ (F Ctrl), 16 (F ELA), 15 (M Ctrl), 16 (M ELA). **e** Representative images of CeAL immunostained with c-Fos+ in

CRF-IRES-Cre;Ai14$^{tdT}$ mice. **f** c-Fos + /DAPI was elevated in ELA mice following startle testing (2-way-ANOVA $_{rearing}$ $F_{(1, 25)} = 5.41$, $p = 0.02$, $\eta^2 = 0.47$) and was not influenced by sex (2-way-ANOVA $_{sex}$ $F_{(1, 25)} = 0.30$, $p = 0.58$, $\eta^2 = 0.12$) or an interaction (2-way-ANOVA $_{sex \times rearing}$ $F_{(1, 25)} = 1.22$, $p = 0.27$, $\eta^2 = 0.21$). $n = 7$ (F Ctrl), 7 (F ELA), 8 (M Ctrl), 7 (F ELA). **g** c-Fos+CRF + /CRF+ was increased in ELA females ($p = 0.003$), but not males ($p > 0.99$; 2-way-ANOVA $_{sex \times rearing}$ $F_{(1,53)} = 5.75$, $p = 0.02$, $\eta^2 = 0.47$ with Bonferroni's multiple comparisons). $n = 7$ (F Ctrl), 7 (F ELA), 8 (M Ctrl), 7 (F ELA). **h** c-Fos+CRF + /c-Fos+ was not impacted by sex, rearing or an interaction (2-way-ANOVA $_{sex}$ $F_{(1,25)} = 2.13$, $p = 0.15$, $\eta^2 = 0.04$; 2-way-ANOVA $_{rearing}$ $F_{(1,25)} = 1.23$, $p = 0.27$, $\eta^2 = 0.07$; 2-way-ANOVA $_{sex \times rearing}$ $F_{(1, 25)} = 3.81$, $p = 0.06$, $\eta^2 = 0.39$). **I** Average startle was positively correlated with c-Fos+CRF + /c-Fos+ (Linear Regression: $F_{(1, 25)} = 14.83$, $R^2 = 0.37$, $p = 0.0007$). $n = 7$ (F Ctrl), 7 (F ELA), 8 (M Ctrl), 7 (F ELA). Plots depict individual mice and means ± SEM. *< 0.05, **< 0.01, ***< 0.001, ****< 0.0001. Source data are provided as a Source Data file. Figure 1a was created with BioRender.com and released under a Creative Commons Attribution-NonCommercial-NoDerivs 4.0 International license.

recruitment of CRF + cells during cued-fear acquisition may contribute to the consequences of ELA on the engagement of these neurons during subsequent threat exposure in females.

Thus far, we have shown that ELA evoked a stronger startle phenotype in male and female mice in the context of both high proximity and ambiguous threat without impacting cue-potentiation of startle, suggesting somatic hyper-reactivity to threat, but not magnitude of fear. We then found that ELA rearing in female mice, but not male, augmented the percent of CRF + neurons that were recruited during startle testing. Lastly, we demonstrated that the recruitment of CeAL CRF + neurons was associated with increased startle.

## ELA rearing caused sex-specific changes in threat-induced Ca + activity of CeAL CRF + neurons

We next assessed the activity of CeAL CRF + neurons in vivo to evaluate the impact of ELA on activity profiles during high and ambiguous threats. We expressed Cre-dependent jGCaMP7s in the CeAL of CRF-ires-Cre mice and used in vivo fiber photometry to measure calcium activity (Ca +) of CeAL CRF + neurons during startle testing (Fig. 2a, b and Supplementary Fig. 3A). We examined Ca + signals during defined time windows: during the onset of CS + (4 sec), during the full CS + period leading up to the WN (30 sec), during the no cue period leading up to the WN (NA; 30 sec) and during the period immediately following the WN (10 sec).

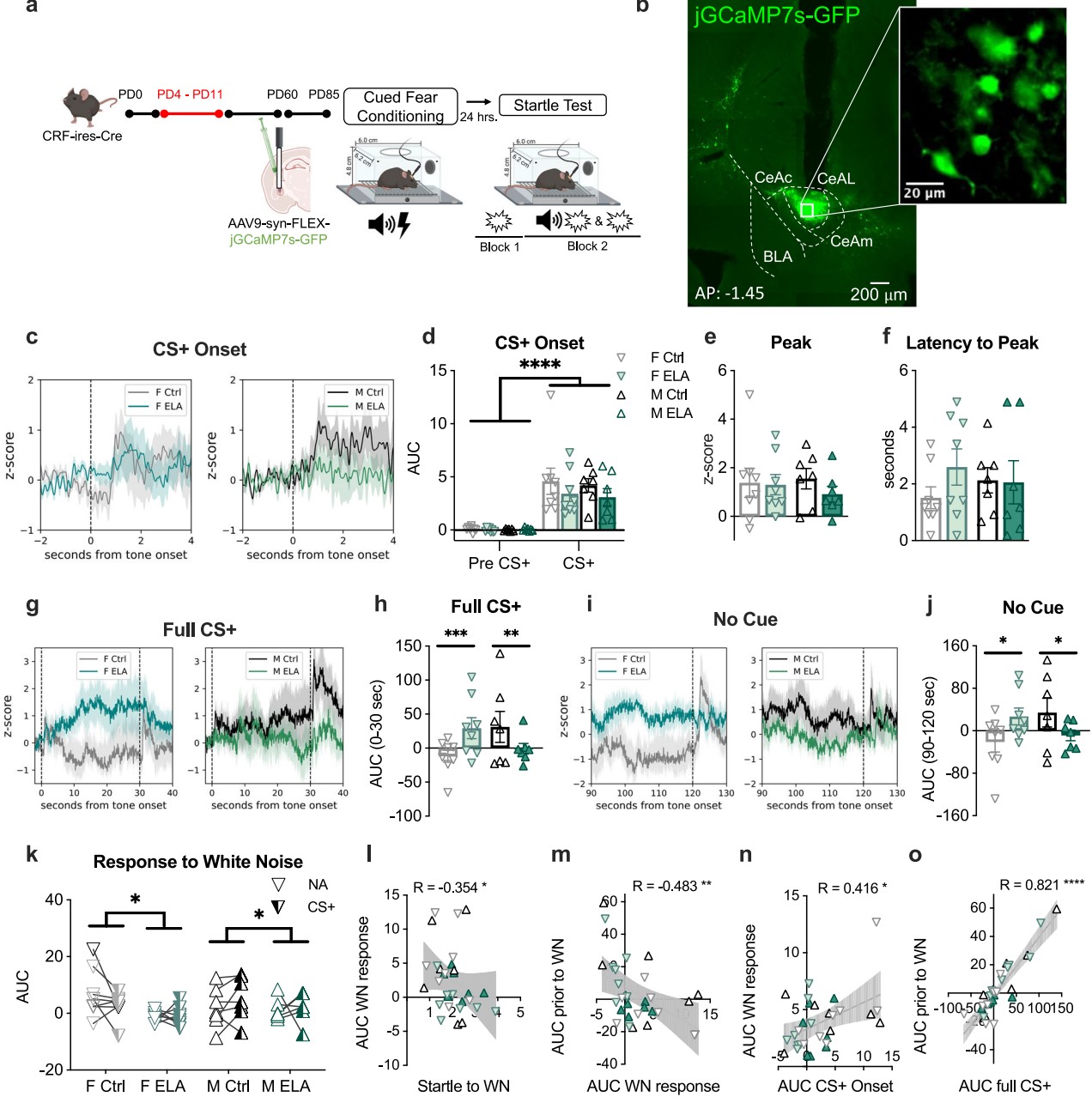

**Fig. 2 | ELA rearing caused sex-specific changes in threat-induced Ca + activity of CeAL CRF + neurons. a** Experimental design. **b** Representative image of viral expression. **c** Activity traces during CS + onset. **d** CRF + neuron activity was elevated during CS + onset (3-way-RM-ANOVA $_{tone}$ $F_{(1,26)} = 68.76$, $p = 8.9e-9$, $\eta^2 = 1.60$; 3-way-RM-ANOVA $_{sex}$ $F_{(1,26)} = 1.14$, $p = 0.70$, $\eta^2 = 0.07$; 3-way-RM-ANOVA $_{rearing}$ $F_{(1,26)} = 0.17$, $p = 0.20$, $\eta^2 = 0.24$; 3-way-RM-ANOVA $_{sex \times rearing}$ $F_{(1,26)} = 0.01$, $p = 0.23$, $\eta^2 = 0.02$). $n = 8$ (F Ctrl, F ELA), 7 (M Ctrl, M ELA). **e** Peak activity was not impacted by rearing or sex (2-way-ANOVA $_{sex}$ $F_{(1,26)} = 0.05$, $p = 0.81$, $\eta^2 = 0.04$; 2-way-ANOVA $_{rearing}$ $F_{(1,26)} = 0.59$, $p = 0.44$, $\eta^2 = 0.11$; 2-way-ANOVA $_{sex \times rearing}$ $F_{(1,26)} = 0.36$, $p = 0.55$, $\eta^2 = 0.10$). $n = 8$ (F Ctrl, F ELA), 7 (M Ctrl, M ELA). **f** Latency to peak z-score was not impacted by rearing or sex (2-way-ANOVA $_{sex}$ $F_{(1,26)} = 0.004$, $p = 0.94$, $\eta^2 = 0.01$; 2-way-ANOVA $_{rearing}$ $F_{(1,26)} = 0.78$, $p = 0.38$, $\eta^2 = 0.14$; 2-way-ANOVA $_{sex \times rearing}$ $F_{(1,26)} = 1.01$, $p = 0.32$, $\eta^2 = 0.19$. $n = 8$ (F Ctrl, F ELA), 7 (M Ctrl, M ELA). **g** Activity traces during CS + ($t = 0$-$30$) and WN ($t = 30$) aligned to $t = 0$. **h** Activity during the CS + was increased in ELA females ($p = 0.0006$) and suppressed in ELA males ($p = 0.002$) compared to Ctrls (3-way-RM-ANOVA $_{rearing \times sex}$ $F_{(1,26)} = 6.06$, $p = 0.02$,

$\eta^2 = 0.62$, Bonferroni's multiple comparisons). $n = 8$ (F Ctrl, F ELA), 7 (M Ctrl, M ELA). **i** Activity traces during noise alone (NA; $t = 120$) aligned to $t = 0$. **j** Activity was enhanced in ELA females ($p = 0.02$) and suppressed in ELA males ($p = 0.04$; 3-way-RM-ANOVA $_{rearing \times sex}$ $F_{(1, 26)} = 5.70$, $p = 0.02$, $\eta^2 = 0.78$, Bonferroni's multiple comparisons). $n = 8$ (F Ctrl, F ELA), 7 (M Ctrl, M ELA). **k** Activity to the WN was decreased by ELA (3-way-RM-ANOVA $_{rearing}$ $F_{(1, 26)} = 5.91$, $p = 0.02$, $\eta^2 = 0.75$, $_{cue}$ $F_{(1, 26)} = 0.003$, $p = 0.95$, $\eta^2 = 0.10$). $n = 8$ (F Ctrl, F ELA), 7 (M Ctrl, M ELA). **l** Startle was negatively correlated with AUC following WN (Two-tailed Spearman $R = -0.354$, $p = 0.05$). **m** AUC prior to WN was negatively correlated with AUC following WN (Two-tailed Spearman $R = -0.483$, $p = 0.006$). **n** AUC to CS + onset was positively correlated with AUC following WN (Two-tailed Spearman $R = 0.416$, $p = 0.02$). **o** AUC during CS + was positively correlated with AUC prior to WN (Two-tailed Spearman $R = 0.821$, $p = 2.8e-8$). Plots depict individual mice and means ± SEM. *< 0.05, **< 0.01, ***< 0.001, ****< 0.0001. Source data are provided as a Source Data file. Figure 2a was created with BioRender.com and released under a Creative Commons Attribution-NonCommercial-NoDerivs 4.0 International license.

To determine if Ca + activity evoked by the onset of the CS + was altered by ELA or sex, we calculated the area under the curve (AUC), peak z-score, and latency to peak z-score during the first 4 sec of the CS +. We observed a significant increase in activity to the CS + compared to activity immediately prior to the CS +, with no evidence for group differences in these measures, indicating comparable patterns of CRF + neuron engagement to threat onset (Fig. 2c–f and Supplementary Fig. 3D). In contrast, activity during the full CS + presentation revealed a sex-dependent impact of ELA, such that Ca + activity was heightened in female ELA mice, compared to female Ctrls, and was attenuated in male ELA mice, compared to male Ctrl (Fig. 2g, h and Supplementary Fig. 3e). To determine if heightened activity observed in female ELA mice in response to the full CS + was attributed to the threat-predictive significance of the CS + or to sensitivity of the tone itself, we examined the Ca + traces in response to the tones during CS acquisition. We found no group differences in Ca + activity following the first tone presentation, indicating the absence of pre-existing sensitivity to the tone itself (Supplementary Fig. 3B). Similarly, group differences were not evident in response to subsequent tone presentations (Supplementary Fig. 3C). Therefore, the increased activity of CeAL CRF + neurons in female ELA mice was not observed during CS acquisition but was specific to threat-induced recall.

To assess whether these group differences in CS-induced activity endured during the no-cue period, we calculated the AUC during $t = 90\text{-}120$ sec using traces that were normalized to CS + onset ($t = 0$ sec). Indeed, these analyses revealed that group differences persisted in the absence of the CS +, such that female ELA mice continued to exhibit elevated Ca + activity compared to female Ctrls, and male ELA mice displayed reduced Ca + activity compared to male Ctrls (Fig. 2l, j and Supplementary Fig. 3F). Next, we explored whether these differences stemmed from sustained activity associated with the CS + or whether changes in activity would emerge independently of CS-induced activity. To do this, we normalized the no-cue traces to $z = 0$ at $t = 90$ and calculated the AUC during $t = 90\text{-}120$ s (Supplementary Fig. 3G). These calculations masked group differences observed prior to normalization, indicating that sustained activity of CeAL CRF + neurons independent of the CS + is not impacted by ELA rearing. Thus, the persistent activity of CeAL CRF + neurons elicited by proximate threat, as opposed to the onset of threat or activity unrelated to the CS +, was altered as a consequence of ELA, with opposing effects in females and males. Example traces can be found in Supplementary Fig. 3H. These data suggest that ELA rearing in females likely impairs the ability to suppress or regulate excessive CRF + neuron activity after the initial threat-induced activation.

We next analyzed Ca + activity aligned with the presentation of the WN and observed reduced activity to the WN in ELA-reared mice (Fig. 2k). Moreover, activity to the WN did not differ by whether the WN co-terminated with the CS + or occurred in the absence of CS +, suggesting that the presence or absence of a threat cue does not influence the response of CRF + neurons to the startling WN Fig. 2k.

To investigate the patterns of CRF + neuron activity that may contribute to heightened startle, we conducted regression analyses examining the relationship between startle and the AUC calculated during the CS + onset, full CS +, and no-cue period (Supplementary Fig. 3I). Unexpectedly, startle to the WN was not correlated with CRF+ neuron activity during the CS + onset, full CS + or the no cue period leading up to WN. These findings indicate that CRF + neurons may not directly drive the motor response per se but may instead contribute to the preparation of downstream systems that engage the motor response[21,48,49]. When we next examined whether neural predictors of startle were present in our measurements, we found that heightened startle was associated with reduced CRF + neuron activity to the WN (Fig. 2l and Supplementary Fig. 3I). Considering this relationship, and in light of evidence for attenuated CRF + neuron activity following WN-induced engagement of flight[22,50], we assessed whether the prior activity state of CRF + neurons

was associated with attenuated activity in response to the WN. Indeed, greater CRF + neuron activity immediately preceding the WN was associated with the attenuated activity to the WN (Fig. 2m and Supplementary Fig. 3I). By contrast, increased activity to CS + onset was predictive of increased activity to the WN (Fig. 2n and Supplementary Fig. 3I). Therefore, the temporal patterns to CRF + neuron activity seem to be critical for heightened startle. Prolonged activity induced by threat may reflect a preparatory function that potentiates startle by priming downstream target neurons for potential threat, subsequently attenuating activity upon startle engagement. Consistent with this, increased CRF + neuron activity lasting the full CS + was positively correlated with greater activity prior to the WN (Fig. 2o and Supplementary Fig. 3I). Collectively, these results suggest that sustained CRF + neuron activity in anticipation of threat or failure to display a temporally precise response to CS + onset and may represent a neural signature for enhance startle.

## CeAL CRF + neurons were necessary for within-session startle but not freezing, with a lasting impact on both startle and freezing

If the sustained activity of CeAL CRF + neurons contributes to heightened startle, then prolonged inhibition of this population of cells should diminish startle. To test this, we conditionally expressed the inhibitory hM4D(Gi) receptor in CRF + neurons by bilaterally injecting AAV9-hSyn-DIO-hM4D(Gi)-mCherry or AAV9-hSyn-DIO-mCherry virus into CeAL of adult CRF-ires-Cre male and female mice that were reared under Ctrl or ELA conditions (Fig. 3a). Following a 3–4-week recovery, mice underwent startle testing 24 h. after cued-fear conditioning. Clozapine N-oxide (CNO) was administered via intraperitoneal (IP) injection at a dose of 1 mg/kg 30 min prior to startle testing such that CeAL CRF + neurons would be persistently silenced throughout the startle testing session.

Startle was significantly reduced in mice that expressed hM4di-mCherry receptor compared to the mCherry control during the NA trials in Block 1 and both NA and CS+ trials in Block 2 (Fig. 3b). To determine if CRF + neuron inhibition was effective across sex and rearing groups, we compared the average startle in Blocks 1 and 2 across experimental conditions. We found that, independent of sex or rearing, hM4di-mCherry expressing mice exhibited a reduction in startle compared to mCherry controls (Fig. 3c), demonstrating that silencing CeAL CRF + neurons attenuates startle across all groups.

Given recent evidence indicating a selective role of CeAL CRF + neurons in the expression of active flight over passive freezing[22], we investigated the impact of CRF + neuron inhibition on freezing during startle testing. Freezing was measured during the habituation period (5 min prior to stimulus presentation), 30 sec leading up to NA trials in Block 1 and 2, and during the 30 sec CS + in Block 2. CeAL CRF + neuron inhibition did not impact freezing (Fig. 3d), confirming the involvement of CeAL CRF + neurons in gating the active expression of startle, and not the passive expression of freezing.

We next addressed whether CeAL CRF + neuron inhibition altered a measure of novelty-induced anxiety by exposing the mice to a 10 min open field test (OFT) 5–7 days after startle testing. Mice received an IP injection of CNO and 30 min later were placed in the open field. Consistent with evidence showing a distinct role of CeAL CRF + neurons on threat-induced anxiety-states over basal anxiety under non-threatening conditions[20,51–53], we did not observe an effect of CeAL CRF + neuron inhibition on the distance traveled, time spent in the center or frequency of center entries in the OFT (Fig. 3f–h and Supplementary Fig. 4A, D–H). Thus, the observed effects of CeAL CRF + neuron inhibition on startle do not appear to be related to motor deficits or alteration in novelty-induced anxiety in the OFT.

Lastly, we performed chemogenetic gain-of-function experiments to address whether augmenting the activity of CeAL CRF + neurons is sufficient to heighten the startle response in Ctrl reared male and female mice (Supplementary Fig. 4I–k). Increasing the activity of CeAL

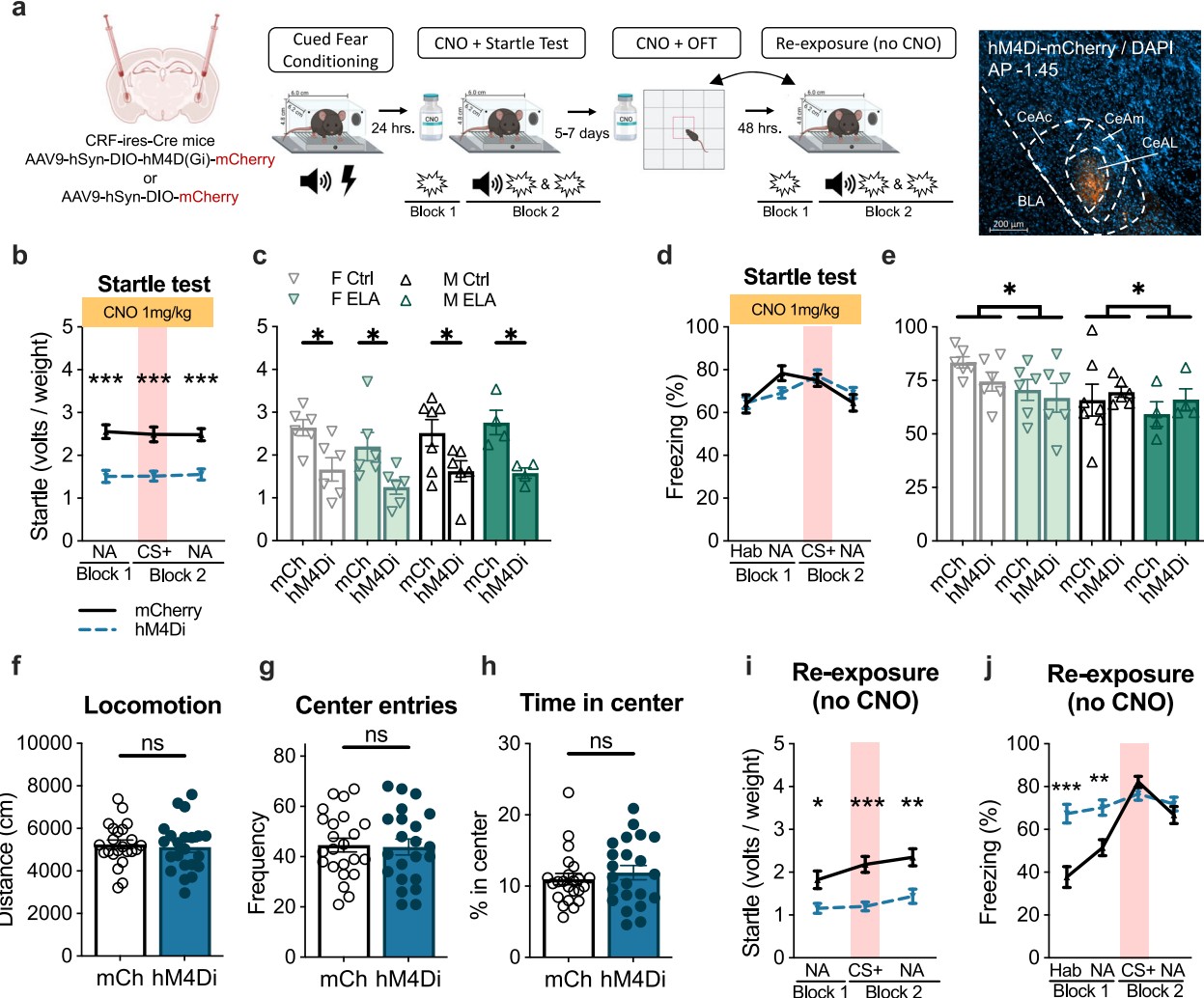

**Fig. 3 | CeAL CRF+ neurons were necessary for within-session startle but not freezing, with a lasting impact on both startle and freezing. a** Experimental design. **b** CeAL CRF + neuron chemogenetic inhibition reduced startle independent of noise alone (NA) or CS + trials (2-way-RM-ANOVA $_{virus}$ $F_{(1, 43)} = 29.42$, $p = 2.4e-6$, $\eta^2 = 1.45$). $n = 23$ (mCh), 22 (hM4di). **c** Average startle (Blocks 1 & 2) was reduced independent of rearing or sex (3-way-ANOVA $_{virus}$ $F_{(1, 37)} = 27.96$, $p = 5.7e-6$, $\eta^2 = 0.86$; $_{rearing}$ $F_{(1, 37)} = 0.75$, $p = 0.39$, $\eta^2 = 0.14$; $_{sex}$ $F_{(1, 37)} = 0.90$, $p = 0.34$, $\eta^2 = 0.15$). $n = 7$ (mCh M Ctrl), $n = 6$ (mCh F Ctrl, hM4Di F Ctrl, mCh F ELA, hM4Di F ELA, hM4Di M Ctrl), $n = 4$ (mCh M ELA, hM4Di M ELA). **d** CeAL CRF + inhibition did not impact percent freezing (2-way-ANOVA $_{virus}$ $F_{(1, 43)} = 0.01$ $p = 0.89$, $\eta^2 = 0.03$). $n = 23$ (mCh), 22 (hM4di). **e** Average freezing was not impacted by CRF + neuron inhibition. Freezing was greater in females and lower in ELA reared mice (3-way-ANOVA $_{virus}$ $F_{(1, 37)} = 0.00009$, $p = 0.97$, $\eta^2 = 0.02$; $_{sex}$ $F_{(1, 37)} = 4.99$, $p = 0.03$, $\eta^2 = 0.36$; $_{rearing}$ $F_{(1, 37)} = 3.86$, $p = 0.05$, $\eta^2 = 0.31$). **f** CRF + neuron inhibition did not impact distance traveled in the open field (Unpaired, Two-Tailed: $t = 0.37$, df = 43, $p = 0.71$, $d = 0.11$).

$n = 7$ (mCh M Ctrl), $n = 6$ (mCh F Ctrl, hM4Di F Ctrl, mCh F ELA, hM4Di F ELA, mCh M Ctrl), $n = 4$ (mCh M ELA, hM4Di M ELA). **g** CRF + neuron inhibition did not impact the frequency of center entries (Unpaired, Two-Tailed: $t = 0.19$, df = 43, $p = 0.84$, $d = 0.05$). $n = 23$ (mCh), 22 (hM4di). **h** CRF + inhibition did not impact the time spent in the center (Unpaired, Two-Tailed: $t = 0.75$, df = 43, $p = 0.45$, $d = 0.10$). $n = 23$ (mCh), 22 (hM4di). **i** Prior CRF + neuron inhibition impaired startle during re-exposure (2-way-RM-ANOVA $_{virus}$ $F_{(1, 43)} = 21.50$, $p = 3.2e-5$, $\eta^2 = 0.96$). $n = 23$ (mCh), 22 (hM4di). **j** Prior CRF + neuron inhibition increased freezing during habituation ($p = 0.0002$) and Block 1 ($p = 0.003$) of re-exposure, but not during CS + ($p = 0.65$; (2-way-RM-ANOVA $_{virus}$ $F_{(1, 43)} = 59.06$, $p = 0.01$, $\eta^2 = 0.70$; $_{cue x virus}$ $F_{(3, 129)} = 26.44$, $p = 2.1e-13$, $\eta^2 = 0.78$). $n = 23$ (mCh), 22 (hM4di). All plots depict individual mice and means ± SEM. *< 0.05, **< 0.01, ***< 0.001. Source data are provided as a Source Data file. Figure 3a created with BioRender.com released under a Creative Commons Attribution-NonCommercial-NoDerivs 4.0 International license.

CRF + neurons did not significantly impact the startle, suggesting that the modulation of the startle is not solely determined by the enhanced activity of these neurons. Rather, it likely involves the integration of signals originating from both local and downstream pathways[48]. In addition, chemogenetically enhancing the activity of CeAL CRF + neurons likely disrupts temporal patterns of activity, which we find may influence the magnitude of startle (Fig. 2l–o).

**Prior CeAL CRF + neuron inhibition caused an enduring suppression of startle**

Previous reports indicate that CRF may act as an important modulatory signal for future experiences[26,27,33,50,54–57]. For example, CRF + neuron

inhibition immediately following cued-fear acquisition has been shown to facilitate the extinction of the fear memory, indicating that the maintenance of CRF + neuron activity is involved in the consolidation or strengthening of a fear memory[33]. Here, we tested whether the attenuated startle caused by CeAL CRF + neuron inhibition would be long-lasting, persisting through re-exposure to startle testing without CNO. After a 5–7 day recovery from initial startle testing (counterbalanced with the OFT), mice underwent a short startle session. We observed that hM4di-mCherry expressing mice, independent of sex and rearing group, continued to exhibit reduced startle compared to the mCherry control group (Fig. 3l and Supplementary Fig. 4G). Interestingly, despite not being necessary for within-session freezing,

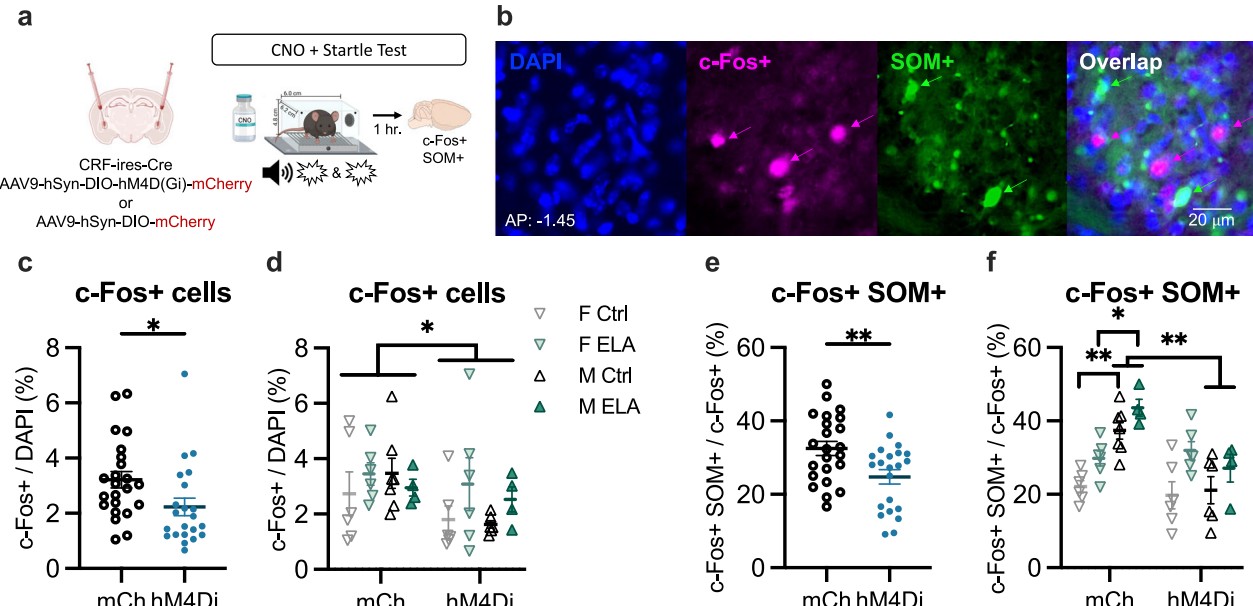

**Fig. 4 | Chemogenetic inhibition of CeAL CRF + neurons reduced CeAL c-Fos +
expression and, in males, reduced activity of SOM + cells. a** Experimental design.
**b** Representative image of CeAL labeled with DAPI, c-Fos +, and SOM + . **c** CeAL
CRF + neuron inhibition during startle testing reduced c-Fos + expression in CeAL
(Unpaired, Two-Tailed: $t = 2.26$, df = 43, $p = 0.02$, $d = 0.67$). $n = 23$ (mCherry: mCh),
22 (hM4di). **d** Suppressed c-Fos+ as a result of CeAL CRF + neuron inhibition
occurred independent of rearing (3-way-ANOVA $_{virus}$ $F_{(1, 37)} = 4.26$, $p = 0.04$,
$\eta^2 = 0.33$; $_{rearing}$ $F_{(1, 37)} = 1.91$, $p = 0.17$, $\eta^2 = 0.27$). $n = 7$ (mCh M Ctrl), $n = 6$ (mCh F Ctrl,
hM4Di F Ctrl, mCh F ELA, hM4Di F ELA, hM4Di M Ctrl), $n = 4$ (mCh M ELA, hM4Di M
ELA). **e** CeAL CRF + neuron inhibition during startle testing reduced c-Fos + co-
expression with SOM + in CeAL (Unpaired, Two-Tailed: $t = 2.83$, df = 43, $p = 0.007$,
$d = 1.25$). $n = 23$ (mCh), 22 (hM4di). **f** Reduced co-expression of c-Fos + with SOM+
following CeAL CRF + neuron inhibition was sex-dependent (3-way-ANOVA $_{virus \times sex}$

$F_{(1, 37)} = 15.82$, $p = 0.0003$, $\eta^2 = 0.65$), occurring in male Ctrl (mCh vs Di: $p = 0.001$)
and male ELA (mCh vs Di: $p = 0.01$) mice but not female Ctrl (mCh vs Di: $p > 0.99$) or
female ELA (mCh vs Di: $p > 0.99$) mice. Further, male mice expressing mCherry
virus showed greater co-expression compared to female mCherry-expressing
mice (M Ctrl vs F Ctrl: $p = 0.002$; M ELA vs F ELA: $p = 0.03$). Lastly, ELA reared
mice exhibited greater co-expression compared to Ctrls (3-way-ANOVA $_{rearing}$
$F_{(1, 37)} = 15.47$, $p = 0.004$, $\eta^2 = 0.64$). $n = 7$ (mCh M Ctrl), $n = 6$ (mCh F Ctrl, hM4Di
F Ctrl, mCh F ELA, hM4Di F ELA, hM4Di M Ctrl), $n = 4$ (mCh M ELA, hM4Di M ELA).
All plots depict individual mice and means ± SEM. *< 0.05, **< 0.01, ***< 0.001.
Source data are provided as a Source Data file. Figure 4a created with
BioRender.com released under a Creative Commons Attribution-NonCommercial-
NoDerivs 4.0 International license.

previous CRF + neuron inhibition increased freezing during Block 1 of
the startle test session but not during the CS + in Block 2 or leading up
to NA trials (Fig. 3j and Supplementary Fig. 4H). Therefore, prior CeAL
CRF + neuron inhibition may have disrupted processes involved in
contextual extinction.

**Chemogenetic inhibition of CeAL CRF + neurons reduced c-
Fos + activity and, in males, reduced activity of SOM + cells**
Our findings demonstrate that CeAL CRF + neuron inhibition attenu-
ates startle without impacting within-session freezing. Further, prior
CRF + neuron inhibition had an enduring impact on fear expression,
diminishing startle a week later and impairing extinction of freezing to
the context. One possible mechanism through which CeAL CRF +
neurons may exert a lasting influence on freezing is by inhibiting local
SOM + neurons[22,32,33,57]. Therefore, we tested whether CeAL CRF +
neuron inhibition altered the activity of local SOM + cells. CNO was
administered 30 min prior to a final short startle test, and brains were
collected an hour following behavior for c-Fos + and SOM + immu-
nostaining (Fig. 4a, b). We first quantified the density of RFP + cells
(RFP + / DAPI) to confirm comparable levels of viral expression in mice
expressing mCherry control virus compared to hM4di virus and the
reduced co-expression of c-Fos in RFP + cells indicating successful
DREADD-mediated inhibition in hM4di-expressing mice (Supplemen-
tary Fig. 4A-C). c-Fos + density was decreased in hM4di-mCherry
expressing mice compared to mCherry mice, independent of sex or
rearing (Fig. 4c, d and Supplementary Fig. 4B), indicating a broad
inhibitory effect of CRF + neuron inhibition on the CeAL. Mice
expressing hM4di-mCherry also showed reduced proportion of c-Fos +
neurons that were SOM + compared to those expressing mCherry

(Fig. 4e). This effect was associated with reduced activity of SOM +
neurons following CRF + neuron inhibition in males, but not females
(Fig. 4f). Moreover, a main effect of rearing revealed overall greater
percent of c-Fos + SOM + cells in ELA reared mice (Fig. 4f). Thus, CRF +
neuronal inhibition led to a sex-dependent alteration in the activity of
SOM + neurons in the CeA, which were also elevated as a result of ELA.

## Discussion

In these experiments, we demonstrate that ELA disrupts typical activity
patterns of CeAL CRF + neurons in response to threat. ELA females
exhibited sustained elevation of activity during conditioned threat and
ELA males showing attenuated patterns of activity, with diminished
activity to the startling WN in both sexes correlating with heightened
startle. We also establish the necessity of CRF + neurons for enhancing
startle and show that prior CRF + neuron inhibition has a prolonged
effect on startle and freezing. Together, our findings provide insight on
the contribution of CeAL CRF + neurons to heightened threat reactivity
following ELA, with implications for understanding sex differences in
symptom susceptibility and manifestation following childhood
trauma.

While we find that CeAL CRF + neurons are necessary for startle,
their activation alone is not sufficient to drive such behavior, indicating
that prolonged activity alone does not fully explain ELA-induced threat
reactivity. Elevated startle responses are likely attributed to dynamic
changes in activation and suppression within the CeAL CRF + neuron
population, rather than absolute levels of neuronal activity. Consistent
with this, the strongest predictor of startle (attenuated activity to the
WN) was correlated with increased activity to CS + onset and
decreased activity prior to the WN, implying that the temporal

dynamics of CeAL CRF + neuron activity may influence the expression of startle. Previous observations indicate that CeAL CRF + neurons exhibit three response profiles: those that selectively respond to the unconditioned stimulus (US), those that selectively respond to the CS and those that transition from US responding to CS responding[33]. This further supports our findings and raises the possibility that aberrant activity or inappropriate distribution of activity profiles of the CeAL CRF + neuron population maybe critical for heightening startle to threat and impacted by ELA.

A key finding from these studies was the effect of ELA on CeAL CRF + neuron activity in females which was observed using c-Fos + immunolabeling and in vivo Ca + imaging. ELA in females led to sustained CS-induced activity of CeAL CRF + neurons during CS+ presentation, persisting in the absence of CS +. ELA in males resulted in attenuated CS-induced activity compared to Ctrl males. In addition, both male and female ELA mice showed diminished activity in response to the WN, correlating with heightened startle. Sex differences in CRF + neuron and CRF receptor 1 (CRFR1) signaling suggest that distinct changes in CRF-mediated synaptic plasticity may underlie startle responses following ELA in males compared to females[58–60]. For instance, disruption of the *Grin1* gene in CeAL CRF + neurons, crucial for NMDA-mediated synaptic plasticity, leads to fear overgeneralization in males but not females[61,62]. In this context, suppressed glutamatergic regulation of CeAL CRF + neurons following ELA may bias males towards a threat-reactive phenotype by disrupting homeostatic regulation[63,64]. Consistently, reports indicate that ELA decreases *Gria2* receptor expression in males, but not females, indicating sex-specific effects on glutamatergic synaptic homeostasis[65,66]. Female ELA mice, on the other hand, maybe more vulnerable to the effects of CRF + hypersecretion on CeAL activity. For example, while prior stress promotes CRF receptor internalization in males, this process is diminished in females[58]. As a result, prior stress in females may result in increased sensitivity to high levels of CRF and inability to adapt to conditions that elicit heightened arousal. While future studies investigating the sex-specific effects of ELA on CRF-mediated synaptic plasticity, our findings offer insights into how ELA distinctly alters the activity of CeAL CRF + neurons in males and females and the relationship between neural activity and threat reactivity.

A methodological limitation to consider is the efficacy of fiber photometry to capture prolonged, tonic-like changes in neuronal activity. While phasic neural activity causes rapid fluctuations in Ca + triggered by discrete cues, sustained changes in activity, particularly during the no-cue period, maybe obscured using this method[67,68]. This is due to the need to standardize signals using z-score calculations for making comparisons across experimental groups. Thus, we cannot exclude the possibility that ELA in males may have led to an overall increase in tonic activity, which may not be adequately detected by the current methods. Studies employing one-photon in vivo Ca + imaging would effectively capture both transient and sustained changes in CeAL CRF + neuron activity with cellular resolution, providing richer insights into the neural dynamics that influence heightened threat reactivity.

Consistent with previous reports, CeAL CRF + neuron inhibition did not reduce time spent in the center of the open field[51–53]. These data indicate that neophobia, anxiety associated with being placed in a novel context, is not dependent on CeAL CRF + neuron activity. Instead, it appears that distinct yet overlapping neural circuitry is involved in heightened vigilance in response to novelty relative to vigilance in response to a learned and imminent threat[52,69]. In line with this, reports indicate that prior immobilization stress is required to observe the anxiolytic effects of CeAL CRF + neuron inhibition in an open field[52]. Therefore, CeAL CRF + neuron-mediated anxiety-like behaviors may require stimulus-driven engagement of CeAL CRF + neurons by external cues.

We also found that CeAL CRF + neurons were necessary for startle but not freezing and that prior chemogenetic inhibition resulted in a lasting suppression of startle and enhanced contextual freezing. These findings indicate that CeAL CRF + neurons gate the engagement of active fear and maybe a locus of plasticity for both active and passive fear expression. A mechanism that supports rapid engagement of startle by CeAL CRF + neurons has been proposed to be by the co-release of GABA, providing simultaneous inhibitory control of CeAL SOM + to suppress freezing[22]. In contrast, CRF neuropeptide release from CeAL CRF + neurons exerts a slower yet long-lasting influence on synaptic function and behavioral states via its actions on CRF receptor 1 (CRFR1), predominantly found on local SOM + neurons[26,61,70–73]. The effects of prior CeAL CRF + neuron silencing on contextual freezing, thus, may reflect the necessity of CRFR1-dependent signaling on SOM + for the induction of contextual fear extinction[26,32,57]. Alternatively, these findings and the observed effects of CeAL CRF + inhibition on the activity of SOM + neurons may in part be due to the co-expression of SOM on CRF + neurons. In addition to co-expressing GABA, CRF-expressing neurons also co-express SOM, neurotensin, and dynorphin which have been shown to support distinct roles in anxiety and fear[52,74,75]. The intricate nature of CeAL CRF signaling by the co-release of neuropeptides, highlights their complex role in orchestrating various behavioral states. Future work in this domain will be critical to understanding how the co-release of neuropeptides regulates behavioral flexibility in response to threat.

Learning to make predictions and rapidly respond to imminent threats is critical for survival but can become maladaptive when inappropriately engaged. The appropriate expression of threat requires a competitive and intricate balance between distinct neuronal subtypes in the CeAL. In this work, we demonstrate that underlying factors of sex and ELA differentially impact the activity of CeAL CRF + neurons to influence threat reactivity, as indexed by startle, and on a competing subpopulation of CeAL neurons, SOM +. Understanding the mechanisms underlying distinct aspects of anxiety and fear have significant translational implications for refining diagnostic criteria and developing more effective treatments. This is particularly crucial given the limited efficacy and mood-related side effects associated with current treatments for fear and anxiety-related disorders[76–78]. It will be critical to address how pre-existing factors may exacerbate or interact with co-occurring symptoms differently in men and women, to optimize treatment development. Collectively, these findings offer valuable insights into the neurobiological mechanisms impacted by sex and ELA and hold promise for personalized treatment strategies that target the underlying root of specific symptoms.

## Methods

### Mice
CRF-ires-Cre (strain B6(Cg)-Crh[tm1(cre)Zjh]/J), JAX stock no. 012704, JAX), Ai14[tdTomato] (strain B6.Cg-Gt(ROSA)26Sor[tm14(CAG-tdTomato)Hze]/J, JAX stock no. 007914) and C57BL/6 N (Charles River) mice were purchased and then bred in house. CRF-ires-Cre female mice were bred with Ai14[tdTomato] male mice to produce CRF-ires-Cre::Ai14[tdT] mice. All mice were maintained on a 12 hr light/dark cycle in a temperature- and humidity- controlled facility (65–79°F, 30–70% respectively) with ab libitum access to standard chow and water. Litters were composed of male and female mice ranging from 3–10 pups per litter. Mice were weaned at postnatal day 21 (PD 21) in sex-segregated groups of 2–4 mice / cage. All experiments were conducted in adult mice (> P70). All animal procedures were approved and conducted in accordance with the guidelines of the Institutional Animal Care and Use Committee of New York State Psychiatric Institute.

### Early life adversity manipulation
Breeding pairs were checked daily for the birth of pups (PD 0) and randomly assigned to early life adversity (ELA) rearing or Ctrl

conditions. The limited bedding and nesting paradigm (LBN) were conducted as previously described[35,79–81]. Briefly, the ELA paradigm consisted of a week period from PD 4 – PD 11 in which the cob bedding was removed and ¾ of the standard cotton nestlet was provided. On PD 11, dams and litter were returned to their standard housing conditions. Dams assigned to the ELA paradigm were maintained as ELA dams for all subsequent litters.

## Viral vectors and coordinates
Surgical procedures were performed on 9–12-week-old CRF-ires-Cre mice. The following viral vectors (Addgene) AAV-Syn-FLEX-jGCaMP7s-WPRE (104491-AAV9), AAV-hSyn-DIO-hM4D(Gi)-mCherry (44362-AAV9), AAV-hSyn-DIO-hM3D(Gq)-mCherry (44361-AAV9), or AAV-hSyn-DIO-mCherry (50459-AAV9) were stereotactically injected (0.3ul) at a rate of 0.1 μl/min using the following CeAL coordinates: AP: − 0.7–1.0 mm, ML:2.6–2.75 mm, DV: − 4.3–4.5 mm. To verify viral expression histologically, mice were deeply anesthetized with 19.5 mL/kg of a 2.5% solution of Avertin (tribromoethanol)- IP) and then transcardially perfused with 4% paraformaldehyde, brains postfixed overnight at 4 °C and cryoprotected in 30% sucrose.

## hM4di-mediated neuronal inhibition
CRF-ires-Cre mice were bilaterally injected with 300 nl of AAV9-hSyn-DIO-hM4D(Gi)-mCherry, AAV9-hSyn-DIO-hM3D(Gq)-mCherry or AAV9-hSyn-DIO-mCherry. Mice recovered for 3-4 weeks. 1 mg/kg of CNO (Tocris CAT# 4936) was administered via IP injection 30 min prior to behavioral sessions. 90 min following the final startle test, mice were deeply anesthetized with 19.5 mL/kg of a 2.5% solution of Avertin (tri-bromoethanol)- IP) and then transcardially perfused with 4% paraformaldehyde, brains postfixed overnight at 4 °C and cryoprotected in 30% sucrose.

## Acoustic startle
Prior to behavior, all mice were transferred from the housing room into an adjacent room for a minimum of 30 min for habituation. Mice were tested for startle in the absence and presence of a conditioned tone. Tone conditioning and startle testing were performed in startle chambers (Med Associates Inc.) equipped with a tone and white noise generator and sensitive load-cell platforms. The load-cell platforms were calibrated for mice (35 grams) using the manufacturer's protocol. The software converts the analog voltage signal from the startle sensor to an arbitrary digital unit (± 2048). The digital startle unit was extracted from the first peak with a minimum value of 50 and converted to volts per the manufacturer's protocol (total startle in arbitrary units / 2048 × 10 volts). To compare startle across mice of different weight, startle in volts was then divided by the animals' weight (volts / weight).

*Conditioning Session:* Following a 5 min habituation to the chamber, mice received 10 tones (30 s, 12 kHz, 70 dB) co-terminating with foot shocks (500 ms, 0.5 mA) with random intertrial intervals of 60–120 sec. The duration of the session was 24 min.

*Startle Test Session:* Startle test sessions were conducted approximately 24 h. after cued-fear conditioning using the manufacturer user's protocol. Following 5 min habituation to the chamber, the startle session consisted of two experimental Blocks: Block 1 was immediately followed by Block 2. Block 1 consisted of 9 white noise (WN) bursts (50 ms; 60 sec ITI) at pseudorandom intensities (100 dB, 105 dB, 110 dB). Block 2 consisted of 18 WN (50 ms; 60 sec ITI) at pseudorandom intensities (100 dB, 105 dB, 110 dB), 9 of which were preceded by the 30 sec conditioned tone (CS +; 12kH, 70 dB). Each mouse underwent identical presentation of stimulus order: Block 1: (105 dB WN, 110 dB WN, 100 dB WN, 110 dB WN, 100 dB WN, 105 dB WN, 100 dB WN, 105 dB WN, 110 dB WN); Block 2 (CS + 105 dB WN, 110 dB WN, CS + 100 dB WN, 100 dB WN, CS + 110 dB WN, 105 dB WN, 110 dB WN, CS + 110 dB WN, 100 dB WN, CS + 105 dB WN, CS + 100 dB

WN, 105 dB WN, CS + 100 dB WN, 105 dB WN, CS + 110 dB WN, 100 dB WN, 110 dB WN, CS + 105 dB WN).

*Short startle test session:* To test for a long-lasting impact of hM4di-mediated neuronal inhibition, DREADD mice were re-exposed to an additional short startle session without clozapine N-oxide (CNO) administration. The short startle session consisted of a 2 min habituation to the chamber and 100 dB WN bursts in the following stimulus presentation: WN, WN, WN, CS + WN, WN, CS + WN, WN, CS + WN. Lastly, to test for the impact of hM4di-mediated inhibition of CeAL CRF + $_+$ on the activity of SOM + neurons, CNO was administered 30 min prior to a final short startle session. Brains were perfused and collected one-hour later for c-Fos + and SOM + immunostaining.

## Open field test
CRF-ires-Cre mice that underwent hM4di-mediated neuronal inhibition were exposed to a 10 min open field test (OFT) 30 min following an interparietal (IP) injection of CNO (Tocris CAT# 4936) at 1 mg/kg. The sessions were recorded and analyzed using Ethovision (Noldus) tracking software.

## Fiber photometry
For in vivo calcium (Ca+) fluorescent recording in awake-behaving mice, a 400 nm diameter fiber-optic cannula was unilaterally lowered over the CeAL of CRF-ires-Cre mice injected with AAV9-Syn-FLEX-jGCaMP7s-WPRE. Mice recovered for 2-3 weeks prior to beginning recording. GCaMP7s signals were recorded using a Tucker Davis Technologies (TDT) LUX RZ10X processor with integrated LEDs and photosensors for 405 nm and 465 nm channels. Signals were captured using a fluorescent MiniCube (Doric Lenses) and acquired using Synapse software (TDT) with LEDs transmitting 30-40uW of light at the fiber tip (measured prior to implantation) and synchronized with behavioral measures using TTL inputs from Med Associates startle chambers. Signals were pre-processed and linked to epoch events using the TDT python package. Raw data from the 405 nm and 465 nm channels were extracted and down sampled by a factor of 10 using a moving window mean. Signal from the 405 nm isosbestic control channel was used to correct for motion artifact and photobleaching by using least-squares linear regression method to fit the 405 nm signal to the 465 nm signal. The change in fluorescence (ΔF/F) was calculated by subtracting the fitted signal from the 465 signal and normalizing to the fitted signal (465 − fitted) / fitted). The ΔF/F were then transformed to z-score values and aligned to the CS + onset ($t = 0$; $z = 0$). AUC data were calculated on the z-scored trace for each time point of interest (CS + onset, full CS +, no cue period, WN response) and then averaged across 9 trials per individual mouse. Ca + activity during the no cue period ($t = 90$–120) was examined in two ways. To assess activity influenced by the CS +, the AUC was calculated using the z-score values aligned to the CS +. To assess activity independent of the CS +, the AUC was calculated using z-score values aligned to $t = 90$, exactly 30 sec prior to the WN. To verify fiber tip location and viral expression histologically, mice were deeply anesthetized with 19.5 mL/kg of a 2.5% solution of Avertin (tribromoethanol)- IP) and then transcardially perfused with 4% paraformaldehyde, brains postfixed overnight at 4 °C and cryoprotected in 30% sucrose. (Supplementary Fig. 3A). For samples in which the slice with the exact fiber tip location was not collected, the tip location was estimated from fiber tracks observed in nearby slices.

## Histology
Mice were deeply anesthetized with 19.5 mL/kg of a 2.5% solution of Avertin (tribromoethanol)- IP) and then transcardially perfused with 4% paraformaldehyde, postfixed overnight at 4 °C and cryoprotected in 30% sucrose. Serial 40 μm coronal sections were sliced using a microtome and stored in cryoprotectant at 20 °C. Immunohisto-chemical procedures were performed on free-floating brain sections.

Sections were incubated in block solution (5% normal goat serum, 5% normal donkey serum, 3% bovine serum albumin, in 0.1% Triton X-100 in 1x PBS) for 2 h. at room temperature (RT), followed by overnight incubation with primary antibodies (rabbit anti-cFos: abcam 190289 or rat anti-SST: Invitrogen MA5-16987) at 4 °C in block solution (1:1000). Slices were then washed 3x for 10 min in 1x PBS and incubated with secondary antibodies (anti-rabbit 647: abcam ab150079, and anti-rat IgG2a FITC conjugate (Thermofisher PA1-84761) for 2 h. at RT in block solution (1:1000). Slices were exposed to DAPI (Thermo Fisher CAT# D1306) for 10 min and washed in 1x PBS (4x, 10 min) before being mounted, coverslipped with PVA-DABCO antifading mounting medium (Sigma Aldrich CAT# 10981) and imaged.

### In situ hybridization (RNAscope)

In situ hybridization of was performed using the ACD Bio RNAscope Multiplex Fluorescent Reagent Kit V2 (Advanced Cell Diagnostics, CAT # 323100), the 4-Plex Ancillary Kit (ACD CAT # 323120) and the associated protocol for freshly frozen tissue. Mice were deeply anesthetized with 19.5 mL/kg of a 2.5% solution of Avertin (tribromoethanol)-IP. Following rapid decapitation, freshly extracted brains were frozen on dry ice and stored at −80 °C. 15 µm sections were collected at −20 °C, mounted on SuperFrost slides (Thermo Fisher Cat# 12-550-15) and stored tightly sealed at −80 °C. Tissue was fixed in 4% paraformaldehyde for 90 min at 4 °C, and then dehydrated at RT in a series of ethanol incubations (50% ETOH, 70% ETOH, 100% ETOH, 100% ETOH; 5 min each). Sections were baked for 15 min at 60 °C before creating a hydrophobic barrier for each slide. Sections were then incubated in hydrogen peroxide (10 min, RT), washed in distilled water (2x, 2 min), incubated in Protease IV (15 min, RT), and washed in 1x PBS (2x, 2 min). Sections were next hybridized in probes for *Crh* (ACD CAT# 316091) and tdTomato (ACD CAT# 317041) for 2 h. at 40 °C and washed in 1x wash buffer (2x, 2 min). The signal was amplified through a series of hybridization steps (AMP 1: 30 min, AMP 2: 30 min, AMP 3: 15 min) at 40 °C and washed in between in wash buffer at RT (2x, 2 min). For fluorescent labeling, sections were incubated in channel-specific HRP for 15 min at 40 °C, washed in 1x wash buffer (2x, 2 min), followed by 30 min incubation with Opal fluorophores at 40 °C (Akoya Biosciences CAT# FP1487001KT; 1:1500 concentration), washed in 1x wash buffer (2x, 2 min), and incubated in HRP blocker at 40 °C for 15 min. Sections were then incubated in ACD Bio DAPI for 2 min, and the coverslipped with ProLong Gold (Thermo Fisher Cat# P36930).

### Statistical analysis

Data were analyzed using Prism 9 (GraphPad Software, Inc.). Normality assessments were performed and followed by appropriate parametric tests with post hoc multiple comparisons with adjusted $p$-values. Mixed-design ANOVAs were used for analyses that contained within-subject variables (e.g., trials, days) and between-subject variables (e.g., rearing condition and sex). Following normality assessment, Pearson coefficients or nonparametric Spearman correlations were computed to establish a relationship between startling reactivity and c-Fos or fiber photometry traces. $p$-values < 0.05 were considered statistically significant. Data are expressed as mean ± SEM.

### Reporting summary

Further information on research design is available in the Nature Portfolio Reporting Summary linked to this article.

## Data availability

Source data are provided in this paper. Experimental design figures were created using BioRender.com. Source data are provided with this paper.

## Code availability

Custom code was not used for the current research.

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

## Acknowledgements

This research was supported by the National Institute of Mental Health F31-MH127888 (C.D.), RO1-MH115914 1098 (K.G.B.) and RO1-MH115049 (K.G.B.). We would also like to acknowledge and thank Dayshalis Ofray and Madalyn Critz, for their support in various technical aspects of the project.

## Author contributions

C.D.: conceptualization, methodology, formal analysis, investigation, data curation, visualization, writing- original draft, writing- review & editing, project administration, funding acquisition; M.P.: investigation, data curation; N.A.: investigation, data curation; G.B.: validation; H.H.: validation, J.B.: validation, B.W.: validation; A.D.: validation; K.G.B.: conceptualization, methodology, writing- review & editing, supervision, project administration, funding acquisition.

## Competing interests

The authors declare no competing interests.

## Ethics and inclusion

The research was conducted following the recommended Inclusion and Ethics policies.
