## [Peer Review File · Nature Communications]

Central amygdala CRF+ neurons promote heightened threat reactivity following early life adversity.REVIEWER COMMENTS

Reviewer #1 (Remarks to the Author):

The manuscript by Demaestri et al. investigated the involvement of CRF-positive neurons in the centrolateral amygdala in fear-potentiated startle. The authors tested the overall hypothesis that the CRF+ population in the CeLAmy mediates the increased fear expression as a result of early life adversity. The study collected data using a variety of techniques, including behavior tests, fiber photometry, chemogenetics and immunostaining, and showed that in transgenic female mice, experience of early life adversity increases the activity of the CRF+ neurons and the startle response to an ambiguous and predictable threat. The fiberphotometry detected male-female differences in the activity of the CRF+ neurons in response to auditory stimuli. Immunostaining for c-Fos confirmed the difference in CRF+ activity and their possible downstream targets, the somatostatin-positive neurons in the central amygdala. Chemogenetic inhibition of the CRF+ population remedy the increased startle response in the female mice but did not affect the mice's freezing behavior in response to CS or their anxiety-like behavior in the open field test. The authors concluded that the early life adversary leads to a greater female susceptibility to anxiety disorders based on the increased activity of the CRF+ neurons in response to threat. Overall, the study is carried out to a high standard, the methods section is well prepared, and the layout of the figures is well designed with descriptive figure legends. However, not all of the data supports the authors' conclusions, and very different interpretations of some of the results are possible. Therefore, several technical issues must be addressed in order to better support the main hypothesis of the paper.

Major concerns:

1. The authors refer to the tdTomato-expressing neurons as CRF+ without confirming that the tdTomato neurons express CRF. There are reported issues with the expression of various markers driven by CRF promoters; therefore, it is essential for the authors to verify the CRF content in the tdTomato-labeled cells in transgenic mice created in house. In addition, the possible changes in CRF/tdTomato colocalization after ELA should be assessed.
2. The CRF+ neurons in the central amygdala are GABAergic neurons that, along with CRF, also coexpress and release GABA, dynorphin, and neurotensin. The fiberphotometry experiment shows activity of the GCaMP7s in CRH-irs-cre transgenic mice, but the CRF content of the recording neurons is again not verified. Knowing the well-established Cre cytotoxicity, the CRF expression by the GCaMP labeled neurons and in the surrounding region should be confirmed. The use of calcium censor does not provide information about what transmitter has been released in the CeAmy upon activation of these neurons; therefore, the experiment does not provide any useful information about the nature of the signaling during the fear-potentiated startle. The use of transmitter-specific biosensors should be applied to verify the release of CRF into the targeted somatostatin neurons.
3. The photometry recordings are time-stamped to the CS tone but not to the mice's behavior. Therefore, there is no correlation between the neuronal activity in CeLAmy and the mouse startle responses. It looks like the neuronal activity picks up a second later than the onset of the tone, so the neurons are activated after the animals' reaction?
4. The ELA did not change the baseline startle or CS response in both sexes of WT mice. (Supplementary Figure 1). It looks like the ELA protocol is ineffective in WT but produces behavior differences only in transgenic mice.
5. The panel on Figure 1E is not informative; the images are of low resolution; the DAPI staining is not helpful at all; the c-Fos expression is indistinguishable at low mag; and there are no images that compare the expression of c-Fos between the groups. A significant increase in c-Fos expression in ELA females should be easily discernible. It is also not shown what percent of tdTomato (CRF+) neurons are colocalized with c-Fos in the CeLAmy of males and females.
6. Similarly, the panel on Figure 4B is uninformative, and the "overlap" image is confusing because it shows colocalization of DAPI, mCherry, c-Fos, and somatostatin. A separate panel showing colocalization of somatostatin neurons with c-Fos would be a much more informative.
7. As the authors point out in the introduction, the role of CRF+ as an anxiogenic factor is well established. Therefore, the inhibition of these neurons that does not affect anxiety-like behavior in the

open-field test casts doubts about the effectiveness of the chemogenetic experiment. The colocalization of the DREADD virus with CRF+ neurons should provide information about the percent of CRF neurons that are inhibited after CNO injection.

Minor:

8. The startle reflex in rodents is triggered by sounds of 85–90 dB intensity and above. It would be very compelling if the increased reactivity of CRF+ neurons in ELA females was confirmed by their startle response to 70-75 dB NA.

Reviewer #2 (Remarks to the Author):

Review of 'Central amygdala CRF+ neurons promote heightened threat reactivity following early life adversity'

This is a potentially important and novel manuscript examining Central amygdala CRF+ neurons as a potential critical mediator of threat response following early life stress (ELA). The authors used a number of tools, including in vivo fiber photometry, to suggest that ELA results in sex-specific changes in the engagement of CeA CRF+ neurons. Chemogenic inhibition of CeA CRF+ selectively diminish startle and produced a potential long-lasting weakening of threat reactivity. Overall the findings are interesting and may suggest sex-specific differences in central amygdala mechanisms of threat and stress responses. However, a number of important concerns lead to limited enthusiasm at this time, with the risk that the data may not support the author's conclusions. Major and minor concerns are outlined below.

Major:

1. In the absence of a statistically significant ANOVA interaction (i.e., sex x ELA/rearing), the authors need to be very careful not to make statements that imply that such an interaction exists. For example, in reference to findings in Fig. 1B on page 3 line 129, the authors state, "It is noteworthy, though, that females may have been more profoundly affected [by early life adversity], evidenced by a significantly higher startle in female ELA mice compared to female Ctrl mice during NA trials in Block 1, NA trials in Block 2, and CS+ trials that did not reach statistical significance in males." In the analysis of these data, the 2-way ANOVA revealed statistically significant main effects of sex and ELA/rearing, but no significant interaction effect is reported, so statistically speaking, there is no difference between male and female sensitivity to ELA as determined by the 2-way ANOVA. In the absence of a significant interaction effect in the ANOVA, pre-planned post-hoc comparisons are inadequate and statistically inappropriate for understanding and describing interactions between the specific levels of different factors and therefore cannot be used to substantiate the interpretation of sex-specific effects of early life trauma on startle. Rather, the significant post-hoc control vs. ELA comparison in females needs to be interpreted in isolation and not as it relates to the non-significant male control vs. ELA comparison.

2. Similarly, in their interpretation of the cFos data, the authors inaccurately point to a sex x ELA/rearing interaction in the absence of a significant ANOVA sex x ELA/rearing interaction effect. There is mention of "a modest sex by rearing interaction" but upon further inspection, the reported interaction is non-significant ($p = 0.06$). Likewise, the interpretation of the correlations depicted in Fig. 1H are misrepresenting the statistical analysis outcome – the presence of a correlation between startle and cFos in females vs. the absence of a similar correlation in males does not evidence a sex x rearing interaction. On the contrary, the absence of a significant 2-way ANOVA interaction in the analysis of these data point to the absence of a sex x rearing interaction. The authors should discuss the data in terms of the significant main effects of sex and ELA/rearing but refrain from discussing interactions in the absence of statistical evidence for an interaction. To ensure accurate data depiction and interpretation, the authors need to rework statements made in lines 148 – 153, the current text in the

Figure 1G and 1H captions, lines 167-176, lines 216-218. It may be worth considering increasing the number of mice per condition in the cFos dataset – reducing the apparent variability in these datasets (Fig. 1F, 1G) may clarify the presence or absence of an interaction. As is, however, there is no statistical grounds for interpreting these data as a reflection of an interaction between sex and rearing condition on cFos activation.

3. It seems like a missed opportunity to not correlate the fiber photometry data with concurrent behavioral startle responses. It would be ideal to show behavioral data synchronized with CS+ and WN presentations along with the experimental and isosbestic channel traces to illustrate any relationships between cell activity and behavior and to establish that the increased fiber photometry signal in ELA females isn't a byproduct of increased movement during ELA-increased startle responding.

4. For the fiber photometry data, it is a bit unclear how the data were pre-processed; specifically, how were data processed using the isosbestic control channel to account for bleaching and potential movement/startle artifacts?

5. The analysis of fiber photometry data during the "no cue" duration needs clarification. As is, the authors provide convincing evidence that ELA causes a sustained increase in cell activity in the absence of the CS during the 30 seconds preceding white noise presentation as shown in Fig 2i at time $t=90$ to $t=120$. This is interpreted (line 233) as a sustained ELA-induced increase in Crh+ CeA cell activity in the absence of CS+. It is somewhat unclear, but the authors then seem to examine the AUC during the same timespan after normalizing trace data such that $z=0$ at $t=90$ for all groups. Then, they analyze these newly normalized AUC data to support the interpretation that signal activity in the absence of prior CS+ is the same across groups (Supp. Fig. 2C). Clearly though, you can eliminate any effect through this same process of normalizing to $z=0$ at a point in time when there is a robust group difference (as there clearly is in control vs. ELA females at $t=90$). It seems that either data handling during this process requires further clarification, or statistical missteps simply masked the effect of ELA on tonic cell activity, leading to the seemingly contradictory interpretations (lines 230-239) of early life adversity effects on cell activity in the absence of the CS+.

6. Regarding the interpretation of fiber photometry data as a demonstration of persistent threat-induced cell activity in ELA females – based on Fig. 2i data showing increased cell activity in ELA females in the absence of the cue, it seems likely that ELA increases tonic Crf+ CeA cell activity in a threat-independent manner. If this is indeed a threat-dependent effect on cell activity, without a non-fear-conditioned ELA group, is there a clear-cut way to determine if the increased cell activity during CS presentation is due to the threat-predictive significance of the CS vs. ELA-heightened sensitivity to the CS tone itself?

7. Regarding data depicted in Fig. 3, the authors should provide full datasets (prior to collapsing across all conditions except virus) either as graphs or tables in either the main figures or supplement.

8. The authors may wish to consider interpreting Figure 4 findings given that 20-35% of CRF CeL cells also express SOM (Kim et al. 2017 <http://dx.doi.org/10.1016/j.neuron.2017.02.034>; McCullough et al. 2018 <http://dx.doi.org/10.1523/ENEURO.0010-18.2018>; Wang et al. eLife 2023;12:e84262; <http://scrnaseq.janelia.org/cea/t2/B4F9B59A-8748-11EE-922D-DB139EC49958/>).

Minor:

1. It would be very helpful if the authors were to add a timeline as part of Figure 2 (like Fig. 1A) to illustrate the fiber photometry experimental design.

2. Second paragraph of introduction, typo on line 59, "valance" should be "valence"

3. Supplementary Figure 1b and 1c Tone 10 data appear to be cut off.

4. Figure 1 caption typos "starte task" and "folowing"

5. Why are the degrees of freedom different for the 2-way ANOVA main effects of sex reported in Fig. 1b [F(1, 53)] vs. Fig. 1d [F(1, 51)]? These analyses were presumably conducted on data collected from the same cohort of mice – did some mice meet an exclusion criterion (e.g., mice that didn't startle during NA Block1)? If so, please describe this in the Statistics subsection of the Methods.

6. The first sentence of Methods subsection on "Startle Test Session" seems to be a bit jumbled.

7. The authors may wish to move the Methods subsections on "Viral Vectors and Coordinates" and "hM4Di-mediated neuronal inhibition" immediately after "Early Life Adversity Manipulation" so that the reader knows the viral approach and CNO administration timeline before discussion of the behavioral protocols in which these approaches were applied (i.e., short startle and open field test).

8. In the "Mice" subsection of the methods, please specify the origin of the mouse line, e.g., Jackson Laboratory and specific stock numbers.

9. Please provide a citation for the mouse LBN ELA protocol (Rice et al. 2008 Endocrinology PMID: 18566122).

Reviewer #3 (Remarks to the Author):

In this interesting and potentially high-impact manuscript, Demaestri and colleagues identify a population of neurons that express corticotropin releasing factor (CRF) in the lateral central amygdala and appear to play a critical role in mediating the effects of early life adversity (ELA) on startle behavior in female mice. c-Fos and photometry studies showed that these neurons appear to be activated in threatening environments and this effect is amplified in female mice exposed to early life adversity. Chemogenetic silencing of these cells reduced startle behavior.

This paper has many strengths. The topic is broadly interesting and timely. There is a pressing need for mechanistic investigations of how stressors in early life influence risk for psychiatric disorders in adulthood -- a relatively understudied question given its clear translational importance. Furthermore, there is strong epidemiological evidence indicating heightened risk for mood and anxiety disorders in women and that some of this risk may be mediated by early life experiences, but the underlying mechanisms are not well understood. This study addresses both gaps in the literature through a series of carefully designed and rigorous experiments.

I have a few suggestions that could further strengthen this work in a revision:

1. In Figure 1, the authors show that startle behavior is enhanced in different contexts in female mice exposed to ELA and this is associated with increased c-Fos expression in CRF-expressing neurons in the amygdala. In Figure 3, they used chemogenetic silencing of these cells to test for a causal link. However, if I am understanding the data presented in Fig. 3c correctly, it looks like there is actually no effect of ELA on startle behavior in female mice (or male mice) in the mCherry control group in this experimental cohort? That is, there is no difference in startle behavior in the female / mCherry / ELA group compared to the female / mCherry / no ELA control group. This seems like it is basically not replicating the finding in Figure 1c, showing that ELA increased startle behavior in essentially the same paradigm? If so, this makes the chemogenetic inhibition data hard to interpret. Is this just due to smaller sample sizes per condition in Figure 3c? If so, perhaps a larger sample would be clarifying. Alternatively, optogenetic inhibition within session might give a clearer answer.

Similarly optogenetic or chemogenetic activation could be useful for bolstering confidence in the claim that increased activity in CRF+ neurons in the BLA is mediating the effect of ELA on startle behavior in females. As is, we have clear evidence that silencing these cells can reduce startle behavior in general in all mice but it's unclear whether the enhanced startle in females exposed to ELA is a robust and reproducible effect and if so, whether that effect is mediated by increased activity in this cell type.

2. In Figure 1, was there a sex-by-ELA interaction indicating a significantly different effect of ELA on startle in females vs. males? Same for Figure 1h: not a big deal because clearly they are different, but again, readers might like to see an interaction statistic.

3. In Figure 2, the authors used photometry to record activity in CRF-expressing amygdala neurons, motivated by the c-Fos studies in Figure 1. Clearly there are differences in activity across the four groups during the CS+ tone presentation as well as during the "no cue" period, which are consistent with the authors' overall claims. However, they did not actually perform any analysis to establish whether there is a significant increase in activity in these cells time-locked to tone onset, unless I missed it. This seems quite important for supporting the claim that activity in these cells is actually responsive to a cue signaling threat, and this would certainly appear to be the case judging qualitatively from the data presented in Figure 2g, at least for female ELA mice. It would be helpful to provide quantitative statistical support for this claim, either in the form of a GLM linking activity in this cell type to CS+ tone presentation or just showing that activity during the tone presentation is higher than the period leading up to the cue onset.

4. "Thus, the observed effects of CeAL CRF++ inhibition on startle do not appear to be related to motor deficits or alteration in anxiety-like behavior." The authors make a strong case that the startle behavior that is the focus of their study is in fact quite relevant for anxiety disorders. Some readers might therefore view startle behavior as an anxiety-related behavior, but here the authors say that the effects are unrelated to anxiety-like behavior. I understand what the authors mean at a high level -- there must be something different about the anxiety-related behavior measured by the open field test and the anxiety-related behavior measured in the startle paradigm. But it might be helpful to include some discussion of this. How are they different? What are the translational implications for our understanding of anxiety disorders, the role of ELA in mediating risk for anxiety disorders in females, and therapeutic strategies based on the findings in this paper?

Conversely, to the extent that startle behavior is something quite distinct from anxiety, what role do the authors think this mechanism might be playing in the pathogenesis of anxiety disorders?

5. Likewise, some additional discussion of the mechanism conferring this sex specific effect might be useful. Additional experimentation establishing a mechanism would of course be optimal but may be outside the scope of this report; discussion might suffice.

6. It's very interesting that the effects of chemogenetic inhibition are so long-lasting. Again, it might be helpful to include some discussion of potential mechanisms (LTD?).

Response to Reviewers

Reviewer #1 (Remarks to the Author):

The manuscript by Demaestri et al. investigated the involvement of CRF-positive neurons in the centrolateral amygdala in fear-potentiated startle. The authors tested the overall hypothesis that the CRF+ population in the CeLAmy mediates the increased fear expression as a result of early life adversity. The study collected data using a variety of techniques, including behavior tests, fiber photometry, chemogenetics and immunostaining, and showed that in transgenic female mice, experience of early life adversity increases the activity of the CRF+ neurons and the startle response to an ambiguous and predictable threat. The fiberphotometry detected male-female differences in the activity of the CRF+ neurons in response to auditory stimuli. Immunostaining for c-Fos confirmed the difference in CRF+ activity and their possible downstream targets, the somatostatin-positive neurons in the central amygdala. Chemogenetic inhibition of the CRF+ population remedy the increased startle response in the female mice but did not affect the mice's freezing behavior in response to CS or their anxiety-like behavior in the open field test. The authors concluded that the early life adversary leads to a greater female susceptibility to anxiety disorders based on the increased activity of the CRF+ neurons in response to threat. Overall, the study is carried out to a high standard, the methods section is well prepared, and the layout of the figures is well designed with descriptive figure legends. However, not all of the data supports the authors' conclusions, and very different interpretations of some of the results are possible. Therefore, several technical issues must be addressed in order to better support the main hypothesis of the paper.

Major concerns:

1. The authors refer to the tdTomato-expressing neurons as CRF+ without confirming that the tdTomato neurons express CRF. There are reported issues with the expression of various markers driven by CRF promoters; therefore, it is essential for the authors to verify the CRF content in the tdTomato-labeled cells in transgenic mice created in house. In addition, the possible changes in CRF/tdTomato colocalization after ELA should be assessed.

We agree with the Reviewer that it is important to verify the expression of *Crf* in the tdTomato-labeled cells in the transgenic mouse bred in house from CRF-ires-Cre (Jax: 012704) crossed with Ai14^{tdTomato} reporter (JAX: 007914). Accordingly, we performed RNAscope in situ hybridization in 4 mice from two independent litters and quantified the co-expression of the endogenous *Crf* transcript with tdTomato (tdT) transcript in the CeAL. In line with previous published observations across several brain regions including the amygdala, hypothalamus, basolateral amygdala, and hippocampus (Walker et al., 2019. Neuropharmacology; Wamsteeker Cusulin et al., 2013. PLoS One; Chen et al., 2015. Endocrinology; Birnie et al., 2023. Nature Communications) we found that the majority of tdT+ cells actively co-expressed *Crf* (76.49%; Figure R1 A-B and Supplementary Figure 2A-B in the revised manuscript). We also quantified the density of endogenous *Crf* transcript expression in 2 Cre negative (Cre-) mice and found comparable density *Crf* (relative to DAPI) in Cre negative (Cre-) and Cre positive (Cre+) mice (Figure R1 C-D and Supplementary Figure 2C-E in the revised manuscript). Thus, the presence of Cre under the *Crf* promoter did not impact *Crf* cell density. These data together validate the use of this transgenic line for investigating ELA and sex effects on the density of CRF-expressing neurons in the CeA. While the reviewer brings up an interesting point regarding colocalization of *Crf*/tdT after ELA, the Cre-loxP system used to express the fluorescent reporter (tdTomato) under the control of the *Crf* promoter is a genetic modification that is largely independent of environmental factors, and therefore is not expected to be significantly impacted by ELA (Tanigushi et al., 2011. Neuron). To partially address this issue, we counted CRF+ neuron density using the tdT reporter in a number of control versus ELA mice (Supplementary Figure 2E). We

did not find any effects of ELA on CRF+ neuron density, suggesting that presence of the reporter did not contribute to a reduction in tdT+ cells. While we did not have tissue from mice to be able to assess overlap of *Crf* transcript with tdT+ neurons directly, these two findings together suggest that ELA does not contribute to a reduction in reporter positive cells. Ongoing experiments will test this more directly as part of a larger study assessing ELA-dependent changes in *Crf* gene expression across several brain regions.

Endogenous *Crf* transcript expression in CRF x Ai14^{tdT} mice using RNAscope

Figure R1: a) Representative image of tdTomato (tdT) transcript and endogenous Crf transcript expression in the CRF-ires-Cre::Ai14^{tdTomato} mouse line using in RNAscope situ hybridization. b) Co-expression of tdT and Crf in tdT cells (red bars) and co-expression of tdT and Crf in Crf cells (green bars). Crf and tdT expression was counted from the left and right hemisphere of 3 CeAL slices (n=6 ROI) in 4 individual mice. The average colocalization of Crf and tdT relative to tdT was 76.49% (Mouse 1: 80.58%, Mouse 2: 67.39%, Mouse 3: 77.35%, Mouse 4: 80.65%). The average colocalization of Crf and tdT relative to Crf was 68.09% (Mouse 1: 57.63%, Mouse 2: 70.05%, Mouse 3: 68.90, Mouse 4: 75.75). c) The density of endogenous Crf transcript expression was quantified in 4 Cre⁺ mice (n=6 slices / mouse; Mouse 1: 19.10%, Mouse 2: 16.07%, Mouse 3: 19.95%, Mouse 4: 16.70%) and in 2 Cre⁻ mice (n=10 slices / mouse; Mouse 1: 16.38%, Mouse 2: 21.13). d) The average density of endogenous Crf transcript expression in the CeAL of Cre⁺ mice (avg: 17.95%) was not different to that of Cre⁻ mice (avg: 18.74%).

2. The CRF+ neurons in the central amygdala are GABAergic neurons that, along with CRF, also coexpress and release GABA, dynorphin, and neurotensin. The fiberphotometry experiment shows

activity of the GCaMP7s in CRH-irs-cre transgenic mice, but the CRF content of the recording neurons is again not verified. Knowing the well-established Cre cytotoxicity, the CRF expression by the GCaMP labeled neurons and in the surrounding region should be confirmed. The use of calcium sensor does not provide information about what transmitter has been released in the CeAmy upon activation of these neurons; therefore, the experiment does not provide any useful information about the nature of the signaling during the fear-potentiated startle. The use of transmitter-specific biosensors should be applied to verify the release of CRF into the targeted somatostatin neurons.

We thank the reviewer for raising these important points. We acknowledge that our methods do not provide information about the neuropeptide released by CRF⁺ neurons. We agree that biosensors would be necessary to make such claims and have accordingly adjusted the description of our results and discussion to clarify that our findings pertain to Ca⁺ activity of CRF⁺ neurons rather than the release of CRF⁺ neuropeptide by the target cells. To address concerns about potential Cre cytotoxicity impacting the endogenous expression of CRF in labeled cells, we performed RNAscope in situ hybridization in mice expressing Cre recombinase under the *Crf* promoter (CRF-ires-Cre) crossed with mice harboring the loxP-flanked-STOP sequence under the CAG promoter (Ai14^{tdTomato}), alongside CRF-ires-Cre⁻ negative (Cre⁻) mice. Our findings revealed significant co-expression of *Crf* and tdT in Cre⁺ mice. Further, *Crf* densities in Cre⁺ mice were comparable to those observed in Cre⁻ mice (Figure R1 and Supplemental Fig 2A-D). These new data confirm the endogenous expression of *Crf* in transgenic mice expressing Cre recombinase under the *Crf* promoter and no appreciable effect on *Crf* cell density relative to Cre⁻ mice. Moreover, we recognize the importance of understanding how transmitter co-release from CRF-expressing neurons contributes to fear-potentiated startle. These experiments have become feasible due to advancements in the development of GRAB sensors, which facilitated the engineering of a CRF sensor published and validated in the PVN last year, well after the current experiments were underway (Wang et al., 2023, Science). The use of these new biosensors in future studies will provide the added precision to measure the release of CRF, as well as other neuropeptides, from CRF neurons in the CeA and the identity of their target neurons.

3. The photometry recordings are time-stamped to the CS tone but not to the mice's behavior. Therefore, there is no correlation between the neuronal activity in CeLAmy and the mouse startle responses. It looks like the neuronal activity picks up a second later than the onset of the tone, so the neurons are activated after the animals' reaction?

We thank the Reviewer for the suggestion to explore time dependent changes in CRF⁺ neuron activity that may be associated with startle. In response, we performed regression analyses between the neural activity during our time points of interest (CS onset, CS full, no cue period, WN response) and startle. These analyses revealed several exciting new points that we believe significantly improves the rigor of our analyses and our understanding of the role of CeAL CRF⁺ neuron activity on startle. The full description of these analyses and associated graphs can be found in the results section, Figure 2L-O, and Supplementary Figure 3I in the revised manuscript. Briefly, we found that attenuated CRF⁺ neuron activity in response to the white noise (WN) was associated with increased startle (Figure 2L) and that activity triggered by the WN was positively correlated with activity to CS⁺ onset, yet negatively correlated with activity during the full CS⁺ (Figure 2L-O; Supplemental Fig. 3I). These new findings highlight that temporal dynamics of CeAL CRF⁺ population activity in response to threat and WN, rather than absolute magnitude of activity at any given time point, may be critical for their role in enhanced startle.

4. The ELA did not change the baseline startle or CS response in both sexes of WT mice.

(Supplementary Figure 1). It looks like the ELA protocol is ineffective in WT but produces behavior differences only in transgenic mice.

We thank the Reviewer for the opportunity to clarify this important point. The ELA manipulation indeed effectively elicited a heightened startle response in both WT and transgenic mice during cue recall, which occurred 24 hours after cued-fear conditioning. These findings are illustrated in Figure 1B-C in WT mice and in Supplemental Figure 2F in transgenic mice. Additionally, the data presented in Supplementary Figure 1A were obtained from a separate group of WT mice before experimental manipulation and demonstrate that the enhanced startle phenotype due to ELA was not evident before cued fear conditioning, indicating that the conditioning session promoted the elevated startle in ELA mice. We have modified the text in the results section to clarify that ELA does not influence baseline startle phenotype but does enhance threat-induced startle in both WT and transgenic mice.

5. The panel on Figure 1E is not informative; the images are of low resolution; the DAPI staining is not helpful at all; the c-Fos expression is indistinguishable at low mag; and there are no images that compare the expression of c-Fos between the groups. A significant increase in c-Fos expression in ELA females should be easily discernible. It is also not shown what percent of tdTomato (CRF+) neurons are colocalized with c-Fos in the CeLAmy of males and females.

We appreciate the suggestion to include representative images that compare the expression of c-Fos between the groups. We have modified Figure 1E to include representative images of c-Fos and CRF::*Ai14^{tdT}* in male and female Ctrl and ELA mice. In addition, the Reviewer's point to include the percent of CRF+ neurons that colocalize with c-Fos+ is well noted. We have included this new data in Figure 1G which revealed a sex by rearing interaction. Post-hoc multiple comparisons indicate that female ELA mice showed a greater proportion of active CRF+ cells compared to female Ctrl, an effect not observed in males.

6. Similarly, the panel on Figure 4B is uninformative, and the "overlap" image is confusing because it shows colocalization of DAPI, mCherry, c-Fos, and somatostatin. A separate panel showing colocalization of somatostatin neurons with c-Fos would be a much more informative.

We have modified the panel in Figure 4B to show co-localization of DAPI, c-Fos+ and SOM+, which we agree is more informative and representative of the data quantified in Figure 4C-F.

7. As the authors point out in the introduction, the role of CRF+ as an anxiogenic factor is well established. Therefore, the inhibition of these neurons that does not affect anxiety-like behavior in the open-field test casts doubts about the effectiveness of the chemogenetic experiment. The colocalization of the DREADD virus with CRF+ neurons should provide information about the percent of CRF neurons that are inhibited after CNO injection.

This is an important observation with regard to our interpretation of our findings. While the activation of CeAL CRF+ has been linked to anxiety-like behavior, there are reports indicating that stress induction prior to testing is essential to observe a reduction in anxiety-like behavior following CRF+ inhibition (Regev et al., 2012. *BiolPsychiatry*; Pomrenze et al., 2019. *J. Neurosci.*). These data indicate that neophobia, or anxiety associated with being placed in a novel context, is not dependent on CeAL CRF+ neuron activity, which is consistent with the cited work. Those studies suggest that distinct neural circuitry is involved in elevations in vigilance in response to novelty (where the threat is ambiguous) relative to anxiety/vigilance in response to a learned and imminent threat. In response, we have expanded our treatment of these points in the discussion

section and have adjusted the word choice in the introduction to clarify the ambiguity between novelty-induced anxiety and stress-induced anxiety. Further, we quantified the density of RFP+ cells in mCherry and hM4Di mice to confirm comparable levels of viral expression between groups and the percent of RFP+ neurons that co-expressed c-Fos, demonstrating the percent of CRF+ neurons that were inhibited following CNO administration. These new panels are found in Supplementary Figure 4B-C.

Minor:

8. The startle reflex in rodents is triggered by sounds of 85–90 dB intensity and above. It would be very compelling if the increased reactivity of CRF+ neurons in ELA females was confirmed by their startle response to 70-75 dB NA.

We share the Reviewer's interest in exploring the extent of the ELA effect on startle reactivity to lower intensity white noise (WN) bursts. Our findings that ELA rearing did not impact baseline startle to 95-105 dB WN tested prior to cued-fear conditioning (Supplemental Figure 1A), but did increase startle following cued-fear conditioning (Fig. 1B-C) indicate that a threat-cue is critical in driving enhanced startle in ELA mice. Further, the enhanced startle phenotype following fear conditioning in ELA female mice was observed independent of the WN intensity tested at 100 dB, 105 dB and 110 dB (Figure R 2B). Moreover, in a separate experimental group, we assessed whether ELA in females impacted the sensitivity of baseline startle to a range of dB intensities (Figure R 2A), which did not reveal changes as a function of rearing condition. While these data do not directly address the Reviewer's suggestion to include WN bursts at 70-75 dB, the evidence provided by the current data do not indicate that we would observe an impact of ELA on baseline startle at a lower intensity nor do we believe that these findings would significantly change our interpretation of the role of CRF+ in driving an enhanced threat-induced startle.

Figure R2: a) Baseline startle response to a series of pseudorandom white noise (WN) bursts at 85 dB, 90 dB, 95 dB, 100 dB, 105 dB, 110 dB, 115 dB, and 120 dB in Ctrl and ELA reared female mice. Startle was influenced by dB intensity (2-way ANOVA dB: $F(4.123, 74.21) = 2.726, p = 0.034$) and was not altered by ELA-rearing (2-way ANOVA rearing: $F(1, 18) = 0.303, p = 0.588$). b) Startle to WN bursts at 100 dB, 105 dB, and 110 dB tested 24 hrs. post fear conditioning in Ctrl and ELA reared female mice. Startle was not influenced by dB intensity (2-way ANOVA dB: $F(1.879, 52.62) = 0.033, p = 0.960$) and was significantly increased

in female ELA mice compared to female Ctrl mice (2-way ANOVA dB: $F(1, 28) = 8.733, p = 0.004$).

Reviewer #2 (Remarks to the Author):

Review of ‘Central amygdala CRF+ neurons promote heightened threat reactivity following early life adversity’

This is a potentially important and novel manuscript examining Central amygdala CRF+ neurons as a potential critical mediator of threat response following early life stress (ELA). The authors used a number of tools, including in vivo fiber photometry, to suggest that ELA results in sex-specific changes in the engagement of CeA CRF+ neurons. Chemogenic inhibition of CeA CRF+ selectively diminish startle and produced a potential long-lasting weakening of threat reactivity. Overall the findings are interesting and may suggest sex-specific differences in central amygdala mechanisms of threat and stress responses. However, a number of important concerns lead to limited enthusiasm at this time, with the risk that the data may not support the author’s conclusions. Major and minor concerns are outlined below.

Major:

1. In the absence of a statistically significant ANOVA interaction (i.e., sex x ELA/rearing), the authors need to be very careful not to make statements that imply that such an interaction exists. For example, in reference to findings in Fig. 1B on page 3 line 129, the authors state, “It is noteworthy, though, that females may have been more profoundly affected [by early life adversity], evidenced by a significantly higher startle in female ELA mice compared to female Ctrl during NA trials in Block 1, NA trials in Block 2, and CS+ trials that did not reach statistical significance in males.” In the analysis of these data, the 2-way ANOVA revealed statistically significant main effects of sex and ELA/rearing, but no significant interaction effect is reported, so statistically speaking, there is no difference between male and female sensitivity to ELA as determined by the 2-way ANOVA. In the absence of a significant interaction effect in the ANOVA, pre-planned post-hoc comparisons are inadequate and statistically inappropriate for understanding and describing interactions between the specific levels of different factors and therefore cannot be used to substantiate the interpretation of sex-specific effects of early life trauma on startle. Rather, the significant post-hoc control vs. ELA comparison in females needs to be interpreted in isolation and not as it relates to the non-significant male control vs. ELA comparison.

Thank you for raising these important statistical concerns. We recognize that the pre-planned post-hoc comparisons following a non-significant interaction are inappropriate for describing potential sex-selective effects in the behavioral data and are separate from the significant main effects. In accordance, we have removed these comparisons and related statements comparing male effects to those observed in females.

2. Similarly, in their interpretation of the cFos data, the authors inaccurately point to a sex x ELA/rearing interaction in the absence of a significant ANOVA sex x ELA/rearing interaction effect. There is mention of “a modest sex by rearing interaction” but upon further inspection, the reported interaction is non-significant ($p = 0.06$). Likewise, the interpretation of the correlations depicted in Fig. 1H are misrepresenting the statistical analysis outcome – the presence of a correlation between startle and cFos in females vs. the absence of a similar correlation in males does not evidence a sex x rearing interaction. On the contrary, the absence of a significant 2-way

ANOVA interaction in the analysis of these data point to the absence of a sex x rearing interaction. The authors should discuss the data in terms of the significant main effects of sex and ELA/rearing but refrain from discussing interactions in the absence of statistical evidence for an interaction. To ensure accurate data depiction and interpretation, the authors need to rework statements made in lines 148 – 153, the current text in the Figure 1G and 1H captions, lines 167-176, lines 216-218. It may be worth considering increasing the number of mice per condition in the cFos dataset – reducing the apparent variability in these datasets (Fig. 1F, 1G) may clarify the presence or absence of an interaction. As is, however, there is no statistical grounds for interpreting these data as a reflection of an interaction between sex and rearing condition on cFos activation.

We appreciate the Reviewer's feedback regarding areas where our statistical methods could be improved. In response, we have adjusted the results section to avoid discussing comparisons in the absence of significant interactions. Additionally, we acknowledge the limitation in the variability of the cFos dataset for accurately depicting or interpreting sex-specific effects. However, initial sample size was based on a-priori power estimates. We are reluctant to add mice to the current group in pursuit of achieving a significant p-value for the interaction and instead have updated the text to eliminate discussion of the "modest" effect and will use these new data to estimate sample sizes for ongoing studies. Accordingly, we have updated the correlation in Figure 1H to represent the data collapsed across groups.

3. It seems like a missed opportunity to not correlate the fiber photometry data with concurrent behavioral startle responses. It would be ideal to show behavioral data synchronized with CS+ and WN presentations along with the experimental and isosbestic channel traces to illustrate any relationships between cell activity and behavior and to establish that the increased fiber photometry signal in ELA females isn't a byproduct of increased movement during ELA-increased startle responding.

We appreciate this suggestion and the opportunity to incorporate relevant correlation analysis into the revised manuscript. In response, we have included an additional paragraph in the results section detailing regression analyses between fiber photometry data and startle responses, as well as associated figures in panels for Figure 2L-O and Supplementary Figure 3I. We believe these analyses significantly improve our understanding of the relationship between CRF+ activity and startle, as well as the potential mechanisms underlying the impact of CRF+ activity on heightened startle. For instance, we found that while CRF+ activity during CS+ onset or full CS+ do not correlate with the motor response, attenuated activity in response to the WN predicted the degree of startle. We also found that temporal activity patterns (such as activity at CS+ onset vs sustained CS-induced activity) exert differing influences on the CRF+ activity state preceding white noise (WN) presentation. Considering that the activity state preceding the WN correlates with the degree of CRF attenuation induced by the WN, these new findings suggest that temporal dynamics of CeAL CRF+ population activity in response to threat and WN, rather than absolute magnitude of activity at any given time point, may be critical for their role in enhancing startle. Further, we have moved the signal traces to the WN presentation and integrated the associated data into the main figure (Figure 1C, D, K), data that was previously included in supplementary material. These data show attenuated CRF+ neuron activity in ELA reared mice during the WN presentation, rather than an increase, addressing concerns regarding potential confounding effects of heightened movement during startle on fiber photometry signal. It is important to note that the traces were pre-processed to ensure that any motion-induced artifacts (e.g., signal on the isosbestic channel) were appropriately removed from the experimental channel. We have expanded the description in the methods section to provide additional details on how the isosbestic channel was used for motion artifact correction.

4. For the fiber photometry data, it is a bit unclear how the data were pre-processed; specifically, how were data processed using the isosbestic control channel to account for bleaching and potential movement/startle artifacts?

We apologize for this oversight. We have modified the methods section to include further methodological details clarifying how the data were pre-processed to control for movement artifacts and photobleaching.

5. The analysis of fiber photometry data during the “no cue” duration needs clarification. As is, the authors provide convincing evidence that ELA causes a sustained increase in cell activity in the absence of the CS during the 30 seconds preceding white noise presentation as shown in Fig 2i at time $t=90$ to $t=120$. This is interpreted (line 233) as a sustained ELA-induced increase in Crh+ CeA cell activity in the absence of CS+. It is somewhat unclear, but the authors then seem to examine the AUC during the same timespan after normalizing trace data such that $z=0$ at $t=90$ for all groups. Then, they analyze these newly normalized AUC data to support the interpretation that signal activity in the absence of prior CS+ is the same across groups (Supp. Fig. 2C). Clearly though, you can eliminate any effect through this same process of normalizing to $z=0$ at a point in time when there is a robust group difference (as there clearly is in control vs. ELA females at $t=90$). It seems that either data handling during this process requires further clarification, or statistical missteps simply masked the effect of ELA on tonic cell activity, leading to the seemingly contradictory interpretations (lines 230-239) of early life adversity effects on cell activity in the absence of the CS+.

Thank you for bringing up this important concern and we apologize for the ambiguity in our data processing. The Reviewer is correct in pointing out that differences across groups were eliminated through the process of calculating the no cue AUC after normalizing the traces from $t=90-120$ to $z=0$. In the initial analyses, we aimed to assess differences between groups during the no cue period that may be influenced by the CS+. Thus, we first calculated the AUC using traces that were z-scored to the CS+ onset (Figure 2I-J), revealing a sex x rearing interaction indicating heightened activity in female ELA mice and attenuated activity in male ELA mice, compared to their control counterparts. Next, we explored whether these differences stemmed from sustained activity associated with the CS+ or whether changes would emerge independently of CS-induced activity. To do this, we normalized the no cue traces to $z=0$ at $t=90$ and calculated the AUC during $t=90-120$ seconds (Supplementary Figure 3G). These calculations masked group differences, suggesting similar levels of tonic activity across groups that was independent of the CS+. However, it's crucial to acknowledge an inherent limitation of fiber photometry recordings and data processing in interpreting these findings. Firstly, fiber photometry primarily captures rapid changes in Ca+ levels associated with neuronal activity, typically triggered by discrete cues (such as CS+, shock, WN, etc). Consequently, the technique may not effectively capture prolonged or tonic shifts in neuronal firing rates, limiting its ability to detect changes in activity in the absence of a cue. Secondly, to compare data across different experimental groups, the data need to be standardized through z-score calculations. While this normalization aids in making comparisons between groups, it may inadvertently obscure subtle variations in tonic activity, thereby limiting our ability to accurately detect small but meaningful differences in baseline neural activity. Therefore, while both AUC calculations provide valuable insights, we have decided that interpreting the no cue period data calculated using traces normalized to the CS+ is most accurate for testing for group differences in CRF+ activity that may result from sustained alterations in neuronal activity following the CS+. In response to the Reviewer's concern, we have revised the methods section to include additional methodological details clarifying how the data were pre-processed and how the AUC during the no cue period was calculated. Additionally, we have further elaborated on our interpretation of the data and highlighted the associated limitations in the discussion section of the revised manuscript.

6. Regarding the interpretation of fiber photometry data as a demonstration of persistent threat-induced cell activity in ELA females – based on Fig. 2i data showing increased cell activity in ELA females in the absence of the cue, it seems likely that ELA increases tonic Crf+ CeA cell activity in a threat-independent manner. If this is indeed a threat-dependent effect on cell activity, without a non-fear-conditioned ELA group, is there a clear-cut way to determine if the increased cell activity during CS presentation is due to the threat-predictive significance of the CS vs. ELA-heightened sensitivity to the CS tone itself?

As addressed in the response to a similar concern in the above paragraph, the technical limitations associated with fiber photometry recordings limits our ability to detect if tonic CRF+ activity is increased in female ELA mice in a threat-independent manner. However, one way to determine if increased activity to the CS+ is due to the threat-predictive significance of the CS vs sensitivity to the CS itself without the additional non-fear conditioned ELA group, is in the fiber photometry data collected from the conditioning session. We analyzed the AUC for Ca+ traces during the first tone presentation (prior to administration of the first shock, t=0-30 sec) that were normalized to tone onset (Figure R3). We did not observe group differences in Ca+ to the first CS presentation, indicating the absence of pre-existing sensitivity to the tone cue itself. Group differences in Ca+ activity were also not apparent to the subsequent CS presentations. These findings indicate that CRF+ activity during the acquisition of the CS+ were not influenced by ELA and did not differ by sex. Thus, changes in CRF+ activity as a result of ELA appear to be specific to threat-induced recall. In response, we have included the additional data in the Supplemental Figure 3B-C.

Figure R3: a) CRF+ CeAL activity in response to the first tone presentation during CS-acquisition. The AUC was calculated on the z-score traces during the 30 sec tone presentations aligned to tone onset. Activity was not influenced by ELA-rearing and did not differ by sex (3-way ANOVA rearing: 3-way ANOVA rearing: $F(1, 30) = 0.09, p = 0.766$; 3-way ANOVA sex: $F(1, 30) = 1.579, p = 0.218$). b) CRF+ CeAL activity in response to subsequent 10 tone presentations during CS-acquisition was not impacted by ELA or sex (3-way ANOVA rearing: 3-way ANOVA rearing: $F(1, 30) = 0.064, p = 0.801$; 3-way ANOVA sex: $F(1, 30) = 1.122, p = 0.298$).

7. Regarding data depicted in Fig. 3, the authors should provide full datasets (prior to collapsing across all conditions except virus) either as graphs or tables in either the main figures or supplement.

We apologize for this oversight. As suggested by the Reviewer, we have now included full datasets prior to collapsing across experimental conditions for the open field data, and the startle and freezing data during re-exposure without CNO in Supplemental Figure 4D-K.

8. The authors may wish to consider interpreting Figure 4 findings given that 20-35% of CRF CeL cells also express SOM (Kim et al. 2017 <http://dx.doi.org/10.1016/j.neuron.2017.02.034>; McCullough et al. 2018 <http://dx.doi.org/10.1523/ENEURO.0010-18.2018>; Wang et al. eLife 2023;12:e84262; <http://scrnaseq.janelia.org/cea/t2/B4F9B59A-8748-11EE-922D-DB139EC49958/>).

Thank you for this suggestion. We have included these citations and discussion of the possibility that our observed effects of CeAL CRF+ neuron inhibition on reduced cFos activity of SOM+ neurons may in part be due to the co-expression of CRF with SOM. We have also included further discussion regarding the implications of neuropeptide co-release from CeAL CRF+ neurons on behavioral flexibility.

Minor:

1. It would be very helpful if the authors were to add a timeline as part of Figure 2 (like Fig. 1A) to illustrate the fiber photometry experimental design.

As suggested, we have included a timeline as part of Figure 2.

2. Second paragraph of introduction, typo on line 59, “valance” should be “valence”

Thank you- this typo has been fixed.

3. Supplementary Figure 1b and 1c Tone 10 data appear to be cut off.

Thank you for catching this. We have corrected the graphs in Supplementary Figure 1b and 1c.

4. Figure 1 caption typos “starte task” and “folowing”

Thank you- these typos have been fixed.

5. Why are the degrees of freedom different for the 2-way ANOVA main effects of sex reported in Fig. 1b [F(1, 53)] vs. Fig. 1d [F(1, 51)]? These analyses were presumably conducted on data collected from the same cohort of mice – did some mice meet an exclusion criterion (e.g., mice that didn’t startle during NA Block1)? If so, please describe this in the Statistics subsection of the Methods.

Thank you for catching this discrepancy. Indeed, the analyses were conducted on data collected from the same cohort of mice. The correct degrees of freedom for these data are (1,53) and have now been corrected in Figure 1d legend.

6. The first sentence of Methods subsection on “Startle Test Session” seems to be a bit jumbled.

We appreciate this suggestion. We have edited the sentence accordingly.

7. The authors may wish to move the Methods subsections on “Viral Vectors and Coordinates” and “hM4Di-mediated neuronal inhibition” immediately after “Early Life Adversity Manipulation” so that the reader knows the viral approach and CNO administration timeline before discussion of the behavioral protocols in which these approaches were applied (i.e., short startle and open field test).

Thank you for this suggestion, we have moved the above subsections immediately after “Early Life Adversity Manipulation”.

8. In the “Mice” subsection of the methods, please specify the origin of the mouse line, e.g., Jackson Laboratory and specific stock numbers.

We have now specified the origin of the mouse line and have included specific stock numbers.

9. Please provide a citation for the mouse LBN ELA protocol (Rice et al. 2008 Endocrinology PMID: 18566122).

We apologize for the oversight. We have added the Rice et al. 2008 citation for the mouse LBN protocol.

Reviewer #3 (Remarks to the Author):

In this interesting and potentially high-impact manuscript, Demaestri and colleagues identify a population of neurons that express corticotropin releasing factor (CRF) in the lateral central amygdala and appear to play a critical role in mediating the effects of early life adversity (ELA) on startle behavior in female mice. c-Fos and photometry studies showed that these neurons appear to be activated in threatening environments and this effect is amplified in female mice exposed to early life adversity. Chemogenetic silencing of these cells reduced startle behavior.

This paper has many strengths. The topic is broadly interesting and timely. There is a pressing need for mechanistic investigations of how stressors in early life influence risk for psychiatric disorders in adulthood -- a relatively understudied question given its clear translational importance. Furthermore, there is strong epidemiological evidence indicating heightened risk for mood and anxiety disorders in women and that some of this risk may be mediated by early life experiences, but the underlying mechanisms are not well understood. This study addresses both gaps in the literature through a series of carefully designed and rigorous experiments.

I have a few suggestions that could further strengthen this work in a revision:

1. In Figure 1, the authors show that startle behavior is enhanced in different contexts in female mice exposed to ELA and this is associated with increased c-Fos expression in CRF-expressing neurons in the amygdala. In Figure 3, they used chemogenetic silencing of these cells to test for a causal link. However, if I am understanding the data presented in Fig. 3c correctly, it looks like there is actually no effect of ELA on startle behavior in female mice (or male mice) in the mCherry control group in this experimental cohort? That is, there is no difference in startle behavior in the female / mCherry / ELA group compared to the female / mCherry / no ELA control group. This seems like it is basically not replicating the finding in Figure 1c, showing that ELA increased startle behavior in essentially the same paradigm? If so, this makes the chemogenetic inhibition data hard to interpret. Is this just due to smaller sample sizes per condition in Figure 3c?

If so, perhaps a larger sample would be clarifying. Alternatively, optogenetic inhibition within session might give a clearer answer.

The Reviewer is correct. Despite our consistent replication of the ELA-rearing effect across cohorts of WT and CRF-ires-Cre mice (as depicted in Supplemental Figure 2F), we did not observe a statistically significant effect of ELA in the data presented in Fig. 3C. We concur that a larger sample size would provide clarity, as the estimated sample size derived from the WT mice (eta squared 0.559) amounts to a total of $N = 44$ ($n = 11$ per experimental condition) for the mCherry cohort. However, considering that the chemogenetic experiments aimed to assess the necessity of CeAL CRF⁺ neurons for startle response rather than investigating the effects of early life adversity (ELA) or sex on the necessity of CeAL CRF⁺ neurons for startle, we felt that these experiments fell outside of the scope of the current manuscript. Importantly, the chemogenetic inhibition data remain robust and crucial in illustrating the necessity of CeAL CRF⁺ neurons in the startle response, as the startle was significantly reduced in all experimental conditions.

In addition to the larger sample size required to detect ELA-induced startle, another possible factor that may have impacted the mCherry data in Figure 3C could be related to modest levels of stress associated with CNO injection. Additional experiments conducted in Ctrl Female WT mice revealed that stress associated with the injection of saline led to a non-significant trend towards increased startle and, similarly, injection of CNO in Ctrl female WT mice resulted in a significantly elevated startle. In contrast, injection stress in ELA Female WT mice, shown here in the form of CNO injection, did not lead to increased startle. These findings suggest that stress linked to injections might have increased variance in the data thereby impacting our ability to detect ELA-associated changes in this particular sample. Despite this, we were still able to observe significant effects of cellular inhibition, which arguably, should have been harder to observe under these conditions.

Figure R4: A) Average startle in female ELA mice was increased compared to controls in both WT ($p = 0.006$) and CRF-ires-Cre ($p = 0.01$) mouse lines. B) Stress associated with injection of saline ($p = 0.07$) or CNO ($p = 0.002$) increased startle in F Ctrl WT mice. DREADD-mediated inhibition of CeAL CRF⁺ cells significantly decreased startle in F Ctrl CRF-ires-Cre mice (mcherry vs Gi: $p = 0.01$), which was also reduced compared to WT F Ctrl mice injected with CNO (Gi vs WT CNO $p = 0.0005$) and saline (Gi vs WT Sal: $p = 0.01$) but not WT Ctrl ($p = 0.20$). These data indicate that CRF⁺ inhibition reduced startle in F Ctrl mice that was potentiated by injection stress but did not alter startle exhibited by WT F Ctrl mice

without prior injection stress. C) Stress associated with injection of CNO did not alter startle in F ELA WT mice ($p = 0.78$). DREADD-mediated inhibition of CeAL CRF⁺ cells significantly decreased startle in F ELA CRF-ires-Cre mice (mcherry vs Gi: $p = 0.01$), and reduced startle compared to WT F ELA mice without ($p = 0.0001$) and with CNO IP injection ($p = 0.005$). These data indicate that CRF⁺ inhibition reduced startle that was enhanced by ELA.

Alternatively, optogenetic inhibition within session might give a clearer answer.

We share the Reviewer's interest in using optogenetics within session to validate the inhibitory DREADD-mediated suppression in startle. However, exploring the effects of optogenetic inhibition within-session would entail brief periods of CeAL CRF⁺ inhibition interspersed with periods of typical firing patterns. These experiments would introduce complexity in regards to the temporal dynamics of how both activity and inhibition of these neurons influence behavior. Therefore, while investigating the nuances of how brief periods, rather than sustained periods, of CeAL CRF⁺ inhibition influences behavior is of significant interest, these experiments would be a significant undertaking and fall beyond the scope of the current study. Nonetheless, in response to the reviewer, we completed additional studies to test whether prolonged optogenetic inhibition of CeAL CRF⁺ neurons causes a reduction in startle, as we observed using DREADD-mediated inhibition. To do this, we expressed a light-driven chloride pump (eNpHR3.0) in CRF cells in the CeAL of Ctrl mice and silenced these cells throughout the entire startle session (Figure R 5A; eNpHR3.0 - light on group). Two additional groups of mice were included to control for potential effects of light or virus on startle. One group received eYFP control virus and light during startle (eYFP- light on) and the second group received the eNpHR3.0 virus but no light during startle (eNpHR3.0 - light off). Following prolonged optogenetic inhibition of CeAL CRF⁺ neurons, we replicated the reduction in startle elicited by DREADD inhibition. Specifically, eNpHR3.0-expressing mice that received light-mediated CRF⁺ inhibition exhibited decreased startle compared to eYFP-expressing mice and NpHR3.0-expressing mice that did not receive light (Figure R5B). While further validation in ELA-reared mice is necessary, the additional optogenetic experiment supports the DREADD studies presented in the manuscript, provide strong evidence for our central claim of the importance of this population of cells in modulating startle reactivity.

Figure R5 A) Experimental design. Prolonged optogenetic inhibition was conducted using an across-subject on or off design such that CRF⁺ neurons were inhibited throughout the entire session. Mice expressing the eNpHR virus were randomly assigned to a light off or light on group. B) Prolonged optogenetic inhibition of CeAL CRF⁺ neurons significantly reduced startle (1-way ANOVA: $F(2, 21) = 8.94$, $p = 0.001$). Post-hoc comparisons revealed significantly reduced startle in eNpHR3.0-expressing mice that received light-mediated CRF⁺ inhibition compared to those that did not receive light ($p = 0.001$) and compared to those expressing eYFP control virus ($p = 0.01$).

Similarly optogenetic or chemogenetic activation could be useful for bolstering confidence in the claim that increased activity in CRF⁺ neurons in the BLA is mediating the effect of ELA on startle behavior in females. As is, we have clear evidence that silencing these cells can reduce startle behavior in general in all mice but it's unclear whether the enhanced startle in females exposed to ELA is a robust and reproducible effect and if so, whether that effect is mediated by increased activity in this cell type.

We appreciate the suggestion to test whether chemogenetic activation of CeAL CRF⁺ is sufficient to enhance startle. In response, we carried out this experiment in Ctrl reared male and female mice. We found that increasing the activity of CeAL CRF⁺ did not significantly impact startle. These findings indicate that startle is not solely influenced by enhanced activity of CeAL CRF⁺ neurons, but rather involves an integration of signals that is likely carried out by both local and downstream pathways. Moreover, these new data are also in line with the newly incorporated linear regression analyses that identify that temporal patterns of both increased and attenuated activity of CeAL CRF⁺ may be meaningful for the enhancement of startle. We have included the chemogenetic activation data below (Figure R6) and in Supplementary Fig. 4I-K and have elaborated on their implications in the Discussion section.

Figure R6: A) Experimental design: CRF-ires-Cre male and female Ctrl mice received CeAL injections of excitatory DREADD (DIO-hM3DGq), inhibitory DREADD (DIO-hM4DGi) or control (DIO-mCherry) virus. Following recovery, mice were subjected to cued-fear conditioning. CNO was administered via IP injection 30 minutes prior to startle testing to test for the effects of CeAL CRF⁺ neuron inhibition and activation on startle. B) 2-way ANOVA revealed a main effect of virus ($F(2, 45) = 10.57, p = 0.002$) that was driven by decreased startle in hM4di-expressing mice compared to mCherry controls ($p = 0.0001$). Startle in hM3dq-expressing female mice was not significantly different than mCherry controls ($p = 0.15$). C) Similarly, a main effect of virus in male Ctrl mice ($F(2,$

45) = 3.103, $p = 0.05$) revealed decreased startle in hM4di-expressing mice compared to mCherry controls ($p = 0.03$). Startle in hM3dq-expressing male mice was not significantly different than mCherry controls ($p = 0.80$).

2. In Figure 1, was there a sex-by-ELA interaction indicating a significantly different effect of ELA on startle in females vs. males? Same for Figure 1h: not a big deal because clearly they are different, but again, readers might like to see an interaction statistic.

Thank you for these questions and we agree that these interactions statistics are important to include. We have modified Figure 1 legend accordingly.

3. In Figure 2, the authors used photometry to record activity in CRF-expressing amygdala neurons, motivated by the c-Fos studies in Figure 1. Clearly there are differences in activity across the four groups during the CS+ tone presentation as well as during the "no cue" period, which are consistent with the authors' overall claims. However, they did not actually perform any analysis to establish whether there is a significant increase in activity in these cells time-locked to tone onset, unless I missed it. This seems quite important for supporting the claim that activity in these cells is actually responsive to a cue signaling threat, and this would certainly appear to be the case judging qualitatively from the data presented in Figure 2g, at least for female ELA mice. It would be helpful to provide quantitative statistical support for this claim, either in the form of a GLM linking activity in this cell type to CS+ tone presentation or just showing that activity during the tone presentation is higher than the period leading up to the cue onset.

We appreciate the suggestion to incorporate statistical analyses showing that CRF+ neurons are indeed responsive to the tone onset. To address this point, we have included AUC data calculated during the pre-tone period to directly compare to the AUC data at CS+ onset (Figure 2D).

4. "Thus, the observed effects of CeAL CRF++ inhibition on startle do not appear to be related to motor deficits or alteration in anxiety-like behavior." The authors make a strong case that the startle behavior that is the focus of their study is in fact quite relevant for anxiety disorders. Some readers might therefore view startle behavior as an anxiety-related behavior, but here the authors say that the effects are unrelated to anxiety-like behavior. I understand what the authors mean at a high level -- there must be something different about the anxiety-related behavior measured by the open field test and the anxiety-related behavior measured in the startle paradigm. But it might be helpful to include some discussion of this. How are they different? What are the translational implications for our understanding of anxiety disorders, the role of ELA in mediating risk for anxiety disorders in females, and therapeutic strategies based on the findings in this paper? Conversely, to the extent that startle behavior is something quite distinct from anxiety, what role do the authors think this mechanism might be playing in the pathogenesis of anxiety disorders?

We agree with this important point and appreciate the opportunity to clarify our interpretation and expand our discussion on the translational relevance of these findings. We have addressed a similar concern in our response to Reviewer 1 above, point #7. We have modified the results and discussion sections to reflect a distinction between novelty-induced anxiety under non-threatening conditions and threat-induced startle.

5. Likewise, some additional discussion of the mechanism conferring this sex specific effect might be useful. Additional experimentation establishing a mechanism would of course be optimal but may be outside the scope of this report; discussion might suffice.

Thank you for this suggestion. We agree that additional discussion exploring the mechanisms underlying the sex-specific effects provides a more comprehensive interpretation of these results.

We have contextualized our findings with existing literature on potential sex-specific mechanisms that may underlie our sex-specific effects. We also identify areas that would advance future research.

6. It's very interesting that the effects of chemogenetic inhibition are so long-lasting. Again, it might be helpful to include some discussion of potential mechanisms (LTD?).

Thank you for this suggestion. We have included additional discussion of the mechanisms conferring the long-lasting effects of chemogenetic inhibition.

REVIEWERS' COMMENTS

Reviewer #1 (Remarks to the Author):

The authors addressed very well all of my concerns in the revised manuscript.

Reviewer #2 (Remarks to the Author):

The authors have thoughtfully responded to my remaining concerns.

Reviewer #3 (Remarks to the Author):

The authors have done a nice job of responding to all of my comments, including the addition of significant new data. Reading the other reviews, it looks like there was substantial convergence in our comments, and in my opinion, the authors have done a nice job responding to those critiques as well. This manuscript will have an important impact on the field and I look forward to seeing it in print.